# Self-Correcting Bayesian Optimization through Bayesian Active Learning

**Carl Hvarfner**
carl.hvarfner@cs.lth.se
Lund University

**Erik Orm Hellsten**
erik.hellsten@cs.lth.se
Lund University

**Frank Hutter**
fh@cs.uni-freiburg.de
University of Freiburg

**Luigi Nardi**
luigi.nardi@cs.lth.se
Lund University
Stanford University
DBtune

## Abstract

Gaussian processes are the model of choice in Bayesian optimization and active learning. Yet, they are highly dependent on cleverly chosen hyperparameters to reach their full potential, and little effort is devoted to finding good hyperparameters in the literature. We demonstrate the impact of selecting good hyperparameters for GPs and present two acquisition functions that explicitly prioritize hyperparameter learning. Statistical distance-based Active Learning (SAL) considers the average disagreement between samples from the posterior, as measured by a statistical distance. SAL outperforms the state-of-the-art in Bayesian active learning on several test functions. We then introduce Self-Correcting Bayesian Optimization (SCoreBO), which extends SAL to perform Bayesian optimization and active learning simultaneously. SCoreBO learns the model hyperparameters at improved rates compared to vanilla BO, while outperforming the latest Bayesian optimization methods on traditional benchmarks. Moreover, we demonstrate the importance of self-correction on atypical Bayesian optimization tasks.

## 1 Introduction

Bayesian Optimization (BO) is a powerful paradigm for black-box optimization problems, i.e., problems that can only be accessed by pointwise queries. Such problems arise in many applications, ranging from including drug discovery [21] to configuration of combinatorial problem solvers [27, 28], hardware design [14, 43], hyperparameter tuning [11, 30, 33, 52], and robotics [4, 9, 40, 41].

Gaussian processes (GPs) are a popular choice as surrogate models in BO applications. Given the data, the model hyperparameters are typically estimated using either Maximum Likelihood or Maximum a Posteriori estimation (MAP) [49]. Alternatively, a fully Bayesian treatment of the hyperparameters [46, 55] removes the need to choose any single set through Monte Carlo integration. This procedure effectively considers all possible hyperparameter values under the current posterior, thereby accounting for hyperparameter uncertainty. However, the relationship between accurate GP hyperparameter estimation and BO performance has received little attention [3, 7, 58, 69, 71], and active reduction of hyperparameter uncertainty is not an integral part of any prevalent BO acquisition function. In contrast, the field of Bayesian Active Learning (BAL) contains multiple acquisition functions based solely on reducing hyperparameter-induced measures of uncertainty [26, 34, 50], and the broader field of Bayesian Experimental Design [1, 10, 48] revolves around acquisition of data to best learns the model parameters.

37th Conference on Neural Information Processing Systems (NeurIPS 2023).

The importance of the GP hyperparameters in BO is illustrated in Fig. 1, which shows average simple regret over 20 optimization runs of 8-dimensional functions drawn from a Gaussian process prior. The curves correspond to the performance of Expected Improvement with noisy experiments (`NEI`) [36] acquisition function under a fully Bayesian hyperparameter treatment using NUTS [25]. Two prevalent hyperparameter priors, described in detail in App. B.1, as well as the true model hyperparameters, are used. Clearly, good model hyperparameters have substantial impact on BO performance, and BO methods could greatly benefit from estimating the model hyperparameters as accurately as possible. Furthermore, the hyperparameter estimation task can become daunting under complex problem setups, such as non-stationary objectives (spatially varying lengthscales, heteroskedasticity) [6, 13, 16, 56, 64], high-dimensional search spaces [15, 47], and additively decomposable objectives [19, 32]. The complexity of such problems warrants the use of more complex, task-specific surrogate models. In such settings, the success of the optimization may increasingly hinge on the presumed accuracy of the task-specific surrogate.

We proceed in two steps. We first introduce *Statistical distance-based Active Learning* (`SAL`), which improves Bayesian active learning by generalizing previous work [26, 50] and introduces a holistic measure of disagreement between the marginal posterior predictive distribution and each conditional posterior predictive. We consider the hyperparameter-induced disagreement between models in the acquisition function, thereby accelerating the learning of model hyperparameters. We then propose *Self-Correcting Bayesian Optimization* (`SCoreBO`), which builds upon `SAL` by explicitly learning the location of the optimizer in conjunction with model hyperparameters. This achieves accelerated hyperparameter learning and yields improved optimization performance on both conventional and exotic BO tasks. Formally, we make the following contributions:

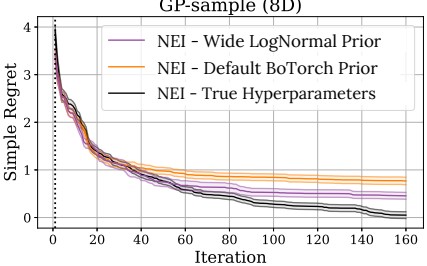

**Figure 1:** Simple regret of using true hyperparameters, BoTorch (v.0.8.4 default) and lognormal hyperparameter priors with fully Bayesian hyperparameter treatment. The prior substantially impacts final performance, and correct hyperparameters yield vastly better results.

1. We introduce `SAL`, a novel and efficient acquisition function for hyperparameter-oriented Bayesian active learning based on statistical distances (Sec. 3.1),
2. We introduce `SCoreBO`, the first acquisition function for joint BO and hyperparameter learning (Sec. 3.2),
3. We display highly competitive performance on an array of conventional AL (Sec. 4.1) and BO tasks (Sec. 4.2), and demonstrate `SCoreBOs`, ability to enhance atypical models such as SAASBO [15] and HEBO [13], and identify decompositions in AddGPs [32](Sec. 4.3).

## 2 Background

### 2.1 Gaussian processes

Gaussian processes (GPs) have become the model class of choice in most BO and active learning applications. They provide a distribution over functions $f \sim \mathcal{GP}(m(\cdot), k(\cdot, \cdot))$ fully defined by the mean function $m(\cdot)$ and the covariance function $k(\cdot, \cdot)$. Under this distribution, the value of the function $f(\boldsymbol{x})$, at a given point $\boldsymbol{x}$, is normally distributed with a closed-form solution for the mean and variance. We assume that observations are perturbed by Gaussian noise, such that $y_{\boldsymbol{x}} = f(\boldsymbol{x}) + \varepsilon, \ \varepsilon \sim N(0, \sigma_\varepsilon^2)$. We also assume the mean function to be constant, such that the dynamics are fully determined by the covariance function $k(\cdot, \cdot)$.

To account for differences in variable importance, each dimension is individually scaled using lengthscale hyperparameters $\ell_i$. For $D$-dimensional inputs $\boldsymbol{x}$ and $\boldsymbol{x}'$, the distance $r(\boldsymbol{x}, \boldsymbol{x}')$ is subsequently computed as $r^2 = \sum_{i=1}^{D} (x_i - x_i')^2 / \ell_i^2$. Along with the outputscale $\sigma_f$, the set $\boldsymbol{\theta} = \{\boldsymbol{\ell}, \sigma_\varepsilon, \sigma_f\}$ comprises the set of hyperparameters that are conventionally learned. The likelihood surface for the GP hyperparameters is typically highly multi-modal [49, 70], where different modes represent different bias-variance trade-offs [49, 50]. To avoid having to choose a single mode, one can define a prior $p(\boldsymbol{\theta})$ and marginalize with respect to the hyperparameters when performing predictions [35]. We outline fully Bayesian hyperparameter treatment in GPs App. G.1.

## 2.2 Bayesian Optimization

Bayesian Optimization (BO) seeks to maximize to a black-box function $f$ over a compact domain $\mathcal{X}$,

$$\boldsymbol{x}^* \in \arg\max_{\boldsymbol{x} \in \mathcal{X}} f(\boldsymbol{x}), \tag{1}$$

such that $f$ can only be sampled point-wise through expensive, noisy evaluations $y_{\boldsymbol{x}} = f(\boldsymbol{x}) + \varepsilon$, where $\varepsilon \sim \mathcal{N}(0, \sigma_\varepsilon^2)$. New configurations are chosen by optimizing an *acquisition function*, which uses the surrogate model to quantify the utility of evaluating new points in the search space. Examples of such heuristics are Expected Improvement (`NEI`) [8, 31] and Upper Confidence Bound (`UCB`) [3, 57, 60]. More sophisticated look-ahead approaches include Knowledge Gradient (`KG`) [17, 68] as well as a class of particular importance for our approach - the information-theoretic acquisition function class. These acquisition functions consider a mutual information objective to select the next query,

$$\alpha_{\texttt{MI}}(\boldsymbol{x}) = I(y_{\boldsymbol{x}}; \ast | \mathcal{D}_n), \tag{2}$$

where $\ast$ can entail either the optimum $\boldsymbol{x}^*$ as in (Predictive) Entropy Search (`ES`/`PES`) [23, 24], the optimal value $f^*$ as in Max-value Entropy Search (`MES`) [42, 59, 65] or the tuple $(\boldsymbol{x}^*, f^*)$, used in Joint Entropy Search (`JES`) [29, 61]. `FITBO` [51] shares similarities with our work, in that the optimal value is governed by a hyperparameter, in their case of a transformed GP.

Within BO, the fully Bayesian hyperparameter treatment is conventionally extended from the predictive posterior to the acquisition function such that for $M$ models with hyperparameters $\boldsymbol{\theta}_m, m \in \{1, \ldots, M\}$ sampled from the posterior over hyperparameters $p(\boldsymbol{\theta}|\mathcal{D})$, the acquisition function $\alpha$ is computed as an expectation over the hyperparameters [46, 55]

$$\alpha(\boldsymbol{x}|\mathcal{D}) = \mathbb{E}_{\boldsymbol{\theta}}[\alpha(\boldsymbol{x}|\boldsymbol{\theta}, \mathcal{D})] \approx \frac{1}{M} \sum_{m=1}^{M} \alpha(\boldsymbol{x}|\boldsymbol{\theta}_m, \mathcal{D}) \quad \boldsymbol{\theta}_m \sim p(\boldsymbol{\theta}|\mathcal{D}). \tag{3}$$

This is also the definition of fully Bayesian treatment considered in this work.

## 2.3 Bayesian Active Learning

In contrast to BO, which aims to find a maximizer to an unknown function, Active Learning (AL) [54] seeks to accurately learn the black-box function globally. Thus, the objective is to minimize the expected prediction loss. AL acquisition functions are classified as either *decision-theoretic*, which minimize the prediction loss over a validation set, or *information-theoretic*, which minimize the space of plausible models given the observed data [26, 37].

In the information-theoretic category, *Active Learning McKay* (`ALM`) [37] selects the point with the highest Shannon Entropy, which for GPs amounts to selecting the point with the highest variance. Under fully Bayesian hyperparameter treatment, it is referred to as Bayesian ALM (`BALM`). *Bayesian Active Learning by Disagreement* (`BALD`) [26] was among the first Bayesian active learning approaches to explicitly focus on learning the model hyperparameters. It approximates the reduction in entropy over the GP hyperparameters from observing a new data point

$$\alpha_{\texttt{BALD}}(\boldsymbol{x}) = I(y_{\boldsymbol{x}}; \boldsymbol{\theta}|\mathcal{D}) = \mathrm{H}(p(y_{\boldsymbol{x}}|\mathcal{D})) - \mathbb{E}_{\boldsymbol{\theta}}[\mathrm{H}(p(y_{\boldsymbol{x}}|\boldsymbol{\theta}, \mathcal{D}))] \tag{4}$$

and was later extended to deep Bayesian active learning [34] and active model (kernel) selection [18]. Lastly, Riis et al. [50] propose a *Bayesian Query-by-Committee* (`BQBC`) strategy. `BQBC` queries where the variance $V$ of the GP mean is the largest, with respect to changing model hyperparameters:

$$\alpha_{BQBC}(\boldsymbol{x}) = V_{\boldsymbol{\theta}}[\mu_{\boldsymbol{\theta}}(\boldsymbol{x}|\mathcal{D})] = \mathbb{E}_{\boldsymbol{\theta}}[(\mu_{\boldsymbol{\theta}}(\boldsymbol{x}|\mathcal{D}) - \mu(\boldsymbol{x}|\mathcal{D}))^2], \tag{5}$$

where $\mu(\boldsymbol{x})$ is the marginal posterior mean at $\boldsymbol{x}$, and $\mu_{\boldsymbol{\theta}}(\boldsymbol{x})$ is the posterior mean conditioned on $\boldsymbol{\theta}$. As such, `BQBC` queries the location which maximizes the average distance between the marginal posterior and the conditionals according to some distance metric (here, the posterior mean), henceforth referred to as hyperparameter-induced *posterior disagreement*. However, disagreement in mean alone does not fully capture hyperparameter-induced disagreement. Thus, [50] also presents *Query-by-Mixture of Gaussian Processes* (`QBMGP`), that adds the `BALM` criterion to the `BQBC` acquisition function.

## 2.4 Statistical Distances

A statistical distance quantifies the distance between two statistical objects. We focus on three (semi-)metrics, which have closed forms for Gaussian random variables. The closed forms expressions, as well as additional intuition on their interaction with Gaussian random variables, can be found in App. G.2.

**The Hellinger distance**   is a dissimilarity measure between two probability distributions which has previously been employed in the context of BO-driven automated model selection by Malkomes et al. [39]. For two probability distributions $p$ and $q$, it is defined as

$$H^2(p, q) = \frac{1}{2} \int_{\mathcal{X}} \left( \sqrt{p(x)} - \sqrt{q(x)} \right)^2 \lambda dx, \tag{6}$$

for some auxiliary measure $\lambda$ under which both $p$ and $q$ are absolutely continuous.

**The Wasserstein distance**   is dissimilarity metric between two distributions describing the average distance one distribution has to be moved to morph into another. The Wasserstein-$k$ distance is defined as

$$W_k(p, q) = \left( \int_0^1 |F_q(x) - F_p(x)|^k dx \right)^{1/k} \tag{7}$$

where, in this work, we focus on the case where $k = 2$.

**The KL divergence**   The KL divergence is a standard asymmetrical measure for dissimilarity between probability distributions. For two probability distributions $P$ and $Q$, it is given by $\mathcal{D}_{KL}(P \parallel Q) = \int_{\mathcal{X}} P(x) \log(P(x)/Q(x)) dx$. The distances in Eq. (6), Eq. (16) and the KL divergence are used for the acquisition functions presented in Sec. 3.

## 3 Methodology

In Sec. 3.1, we introduce `SAL`, a novel family of metrics for BAL. In Sec. 3.2, we extend this to `SCoreBO`, the first acquisition function for joint BO and hyperparameter-oriented active learning, inspired by information-theoretic BO acquisition functions. In Sec. 3.3, we demonstrate how to efficiently approximate different types of statistical distances within the `SAL` context.

### 3.1 Statistical distance-based Active Learning

In active learning for GPs, it is important to efficiently learn the correct model hyperparameters. By measuring where the posterior hyperparameter uncertainty causes high disagreement in model output, the search can be focused on where this uncertainty has a high impact. However, considering only the posterior disagreement in mean, as in `BQBC`, is overly restrictive as it does not fully utilize the available distributions for the hyperparameters. For example, it ignores uncertainty in the outputscale hyperparameter of the Gaussian process, which disincentives exploration. As such, we propose to generalize the acquisition function in Eq. (5) to instead consider the posterior disagreement as measured by any statistical distance. Locations where the posterior distribution changes significantly as a result of model uncertainty are good points to query, in order to quickly learn the model hyperparameters. When an observation at such a location is obtained, hyperparameters which predicted that observation poorly will have a substantially smaller likelihood, which in turn aids hyperparameter convergence. The resulting `SAL` acquisition function is as follows:

$$\alpha_{SAL}(\boldsymbol{x}) = \mathbb{E}_{\boldsymbol{\theta}}[d(p(y_{\boldsymbol{x}}|\boldsymbol{\theta}, \mathcal{D}), p(y_{\boldsymbol{x}}|\mathcal{D}))] \approx \frac{1}{M} \sum_{m=1}^{M} d(p(y_{\boldsymbol{x}}|\boldsymbol{\theta}_m, \mathcal{D}), p(y_{\boldsymbol{x}}|\mathcal{D})), \tag{8}$$

where $M$ is the number of hyperparameter samples drawn from its associated posterior, $\boldsymbol{\theta}_m \sim p(\boldsymbol{\theta}|\mathcal{D})$, $\boldsymbol{\theta} = \{\boldsymbol{\ell}, \sigma_f, \sigma_{\varepsilon}\}$, and $d$ is a statistical distance. Notably, SAL generalizes both BQBC and BALD, which are exactly recovered by choosing the semimetric to the difference in mean or the forward KL divergence, with a short proof for the latter in App. F:

**Proposition 1.** *SAL equipped with the KL-divergence is equivalent to BALD.*

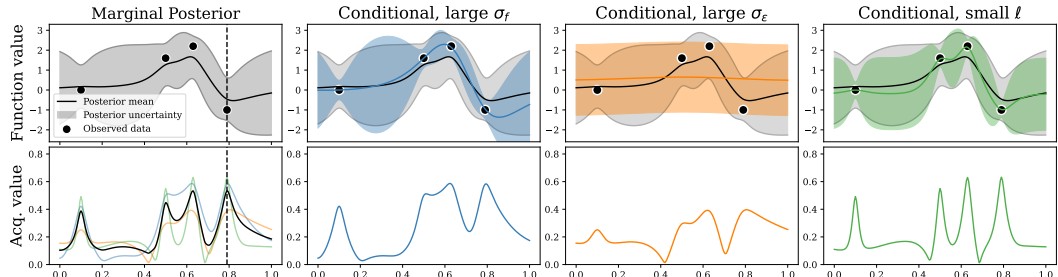

**Figure 2:** Marginal posterior (top left, grey in other plots in top row), $\alpha_{SAL}$ using the Hellinger distance (bottom left, black), and the three conditional GPs (blue, orange, green) and their marginal contribution to the total acquisition function (bottom row). The large disagreement in noise level and lengthscale, primarily caused by the orange GP (large noise, long lengthscale), makes $\alpha_{SAL}$ query the lowest-valued point for a second time (selected location as vertical dashed line in the leftmost plot) to determine the mean and variance at that location.

Fig. 2 visualizes the `SAL` acquisition function. The marginal posterior (left) is made up of three vastly different conditional posteriors with hyperparameters sampled from $p(\boldsymbol{\theta}|\mathcal{D})$ - one with high outputscale (blue), one with very high noise (orange), and one with short lengthscale (green). For each of the blue, orange and green conditionals, the distance to the marginal posterior is computed. Intuitively, disagreement in noise level $\sigma_\varepsilon$ can cause large posterior disagreement at already queried locations. Similarly, uncertainty in outputscale $\sigma_f$ between posteriors will yield disagreement in large-variance regions, which will result in global variance reduction. Compared to other active learning acquisition functions, `SAL` carries distinct advantages: it has incentive to query the same location multiple times to estimate noise levels, and accomplishes the typical active learning objectives of predictive accuracy and global exploration by alleviating uncertainty over the lengthscales and outputscale of the GP. As we show in our experiments (Sec. 4.1, App. D), `SAL` yields superior predictions and reduces hyperparameter uncertainty at drastically improved rates.

## 3.2 Self-Correcting Bayesian Optimization

Equipped with the `SAL` objective from Eq. (8), we have an intuitive measure for the hyperparameter-induced posterior disagreement, which incentivizes hyperparameter learning by querying locations where disagreement is the largest. However, it does not inherently carry an incentive to *optimize* the function. To inject an optimization objective into Eq. (8), we draw inspiration from information-theoretic BO and further condition on samples of the optimum. Conditioning on potential optima yields an additional source of disagreement reserved for promising regions of the search space.

We consider $(\boldsymbol{x}^*, f^*)$, representing the global optimum and optimal value considered in `JES` [29, 61], as hyperparameters. When conditioning on $(\boldsymbol{x}^*, f^*)$, we condition on an additional observation, which displaces the mean and reduces the variance at $\boldsymbol{x}^*$. Moreover, the posterior over $f$ becomes an upper truncated Gaussian, reducing the variance and pushing the mean marginally downwards in uncertain regions far away from the optimum as visualized in Fig. 3. Consequently, sampling and conditioning on $(\boldsymbol{x}^*, f^*)$ introduces an additional source of disagreement between the marginal posterior and the conditionals *globally*. The optimizer $(\boldsymbol{x}^*, f^*)$ is obtained through posterior sampling [67]. For brevity, we hereafter denote $(\boldsymbol{x}^*, f^*)$ by $\divideontimes$. The resulting `SCoreBO` acquisition function is

$$\alpha_{SC}(\boldsymbol{x}) = \mathbb{E}_{\boldsymbol{\theta},\divideontimes}[d(p(y_{\boldsymbol{x}}|\mathcal{D}), p(y_{\boldsymbol{x}}|\boldsymbol{\theta}, \divideontimes, \mathcal{D}))]. \tag{9}$$

The joint posterior $p(\boldsymbol{\theta}, \divideontimes|\mathcal{D}) = p(\divideontimes|\boldsymbol{\theta}, \mathcal{D})p(\boldsymbol{\theta}|\mathcal{D})$ used for the expectation in Eq. (9) can be approximated by hierarchical sampling. We first draw $M$ hyperparameters $\boldsymbol{\theta}$ and thereafter $N$ optimizers $\divideontimes|\boldsymbol{\theta}$. As such, the expression for the `SCoreBO` acquisition function is:

$$\alpha(\boldsymbol{x}) \approx \frac{1}{NM} \sum_{m=1}^{M} \sum_{n=1}^{N} d\left(p(y_{\boldsymbol{x}}|\mathcal{D}), p(y_{\boldsymbol{x}}|\boldsymbol{\theta}_m, \divideontimes_{\boldsymbol{\theta}_{m,n}}, \mathcal{D})\right), \tag{10}$$

where $N$ is the number of optimizers sampled per hyperparameter set. Notably, while the acquisition function in (9) considers the optimizer $(\boldsymbol{x}^*, f^*)$, `SCoreBO` is not restricted to employing that quantity alone. Drawing parallels to `PES` and `MES`, we can also choose to condition on either $\boldsymbol{x}^*$ or $f^*$ alone in place of $(\boldsymbol{x}^*, f^*)$. Doing so introduces a smaller disagreement in the posterior at the conditioned

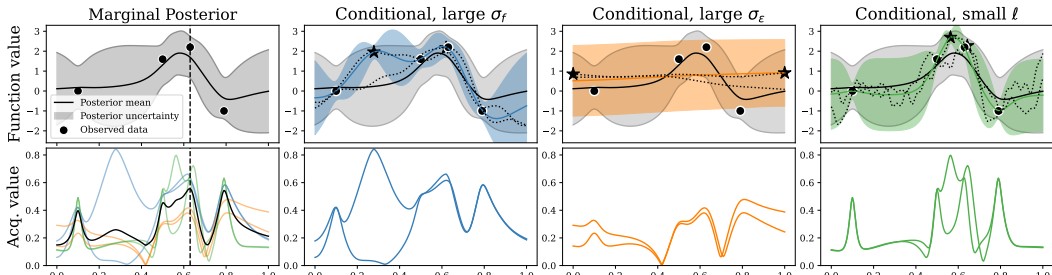

**Figure 3:** Approximate marginal posterior after having conditioned on $(\boldsymbol{x}^*, f^*)$ (top left), $\alpha_{SC}$ using the Hellinger distance (bottom left), the three conditional truncated posteriors and their marginal contribution to the total acquisition function for the same iteration as Fig. 2. Conditioning on $(\boldsymbol{x}^*, f^*)$ (marked as $\star$, drawn from function samples in dashed) inroduces additional disagreement between the marginal posterior and the sampled GPs in promising regions as a result of conditioning. In the figure, we marginalize over $M = 3$ sets of hyperparameters and $N = 2$ optimizers per GP, where each optimizer's contribution to the acquisition function is visible under its corresponding GP. Note that, since function draws are *noiseless*, the conditioned optimum does not need to surpass the best *noisy* observation in value. This phenomenon is most notable in (orange).

---

**Algorithm 1** `SCoreBO` iteration

---

1: **Input:** Number of hyperparameter sets $M$, number of sampled optima $N$, current data $\mathcal{D}$
2: **Output:** Next query location $\boldsymbol{x}'$.
3: **for** $m \in \{1, \ldots, M\}$ **do**
4:     $\boldsymbol{\theta}_m \sim p(\boldsymbol{\theta}|\mathcal{D})$
5:     **for** $n \in \{1, \ldots, N\}$ **do**
6:         $\divideontimes_{\boldsymbol{\theta}_m, n} \leftarrow \max f_{\boldsymbol{\theta}_m, n}$, where $f_{\boldsymbol{\theta}_m, n} \sim p(f|\boldsymbol{\theta}_m, \mathcal{D})$   {Draw $n$ optima for each $\boldsymbol{\theta}_m$}
7:         $p(y_{\boldsymbol{x}}|\boldsymbol{\theta}_m, \divideontimes_{\boldsymbol{\theta}_m, n}, \mathcal{D}) \leftarrow \texttt{CondGP}(\divideontimes_{\boldsymbol{\theta}_m, n}, \boldsymbol{\theta}_m, \mathcal{D})$   {Condition GPs on each optimum}
8:     **end for**
9: **end for**
10: $\boldsymbol{x}' = \arg\max \alpha(\boldsymbol{x})$   {Defined in Eq. (10)}

---

location $\boldsymbol{x}^*$, thus decreasing the acquisition value there. This will in turn decrease the emphasis that `SCoreBO` puts on optimization, relative to hyperparameter learning. In Fig. 3, the `SCoreBO` acquisition function is displayed for the same scenario as in Fig. 2. By conditioning on $N = 2$ optimizers per GP, we obtain $N \times M$ posteriors (displaying the posterior for one out of two optimizers, i.e. the left star in (blue), in Fig. 3). The mean is pushed upwards around the extra observation and the posterior predictive distribution over $f$ is truncated as it is now upper bounded by $f^*$. While the preferred location under `SAL` is still attractive, the best location to query is now one that is more likely to be optimal, but still good under `SAL`.

Algorithm 1 displays how the involved densities are formed for one iteration of `SCoreBO`. For each hyperparameter set, a number of optima are sampled and individually conditioned on (`CondGP`) given the current data and hyperparameter set. After this procedure is completed for all hyperparameter sets, the statistical distance between each conditional posterior and the marginal is computed. The conditioning on the fantasized data point involves a rank-1 update of $\mathcal{O}(n^2)$ of the GP for each draw. As such, the complexity of constructing the acquisition functions is $\mathcal{O}(MNn^2)$ for $M$ models, $N$ optima per model and $n$ data points. We utilize NUTS [25] for the MCMC involved with the fully Bayesian treatment, at a cost of $\mathcal{O}(Dn^3)$ per sample.

### 3.3 Approximation of Statistical Distances

We consider two proper statistical distances, Wasserstein distance and Hellinger distance. In contrast to `BQBC`, the statistical distance between the normally distributed conditionals and the marginal posterior predictive distribution (which is a Gaussian mixture), is not available in closed-form. We propose two approaches: estimating the distances using MC, which we outline for both distances in App E.1, and estimation using moment matching (MM), which we outline below.

**Approximation through Moment Matching**   We propose to fully utilize the closed-form expressions of the involved distances for Gaussians, and approximate the full posterior mixture $p(y_{\boldsymbol{x}}|\mathcal{D})$ with a Gaussian distribution using moment matching (MM) for the first and second moment. While a Gaussian mixture is not generally well approximated by a Normal distribution, we show empirically in App. E that the distance between the conditionals and the approximate posterior is small. In the moment matching approach, the conditional posterior $p(y_{\boldsymbol{x}}|\boldsymbol{\theta}, *, \mathcal{D})$ utilizes a lower bound on the change in the posterior induced by conditioning on $*$, as derived in GIBBON [42], which conveniently involves a second moment matching step of the extended skew Gaussian [45] $p(y_{\boldsymbol{x}}|\boldsymbol{\theta}, *, \mathcal{D})$. This naive approach circumvents a quadratic cost $\mathcal{O}(N^2 M^2)$ in the number of samples of each pass through the acquisition function, and yields comparable performance to the MC estimation procedures proposed in App. E.1. In App. E, we qualitatively assess the accuracy of the MM approach for both distances, and display its ability to preserve the shape of the acquisition function.

# 4   Experiments

In this section we showcase the performance of the `SAL` and `SCoreBO` acquisition functions on a variety of tasks. For active learning, `SAL` shows state-of-the-art performance on a majority of benchmarks, and is more robust than the baselines. For the optimization tasks, `SCoreBO` more efficiently learns the model hyperparameters, and outperforms prominent Bayesian optimization acquisition functions on a variety of tasks. All experiments are implemented in BoTorch [2][1]. We use the same $\mathcal{LN}(0, 3)^2$ hyperparameter priors as Riis et al. [50] unless specified otherwise. `SCoreBO` *and all baselines* utilize fully Bayesian treatment of the hyperparameters. The complete experimental setup is presented in detail in Appendix B, and our code is publicly available at `https://github.com/hvarfner/scorebo.git`. We utilize the moment matching approximation of the statistical distance. Experiments for the MC variant of `SCoreBO` are found in App. E.2.

## 4.1   Active Learning Tasks

To evaluate the performance of `SAL`, we compare it with `BALD`, `BQBC` and `QBMGP` on the same six functions used by Riis et al. [50]: Gramacy (1D) has a periodicity that is hard to distinguish from noise, Higdon and Gramacy (2D) varies in characteristics in different regions, whereas Branin, Hartmann-6 and Ishigami have a generally nonlinear structure. We display both the Wasserstein and Hellinger distance versions of `SAL`, denoted as `SAL-WS` and `SAL-HR`, respectively. We evaluate each method on their predictive power, measured by the negative Marginal Log Likelihood (MLL) of the model predictions over a large set of validation points. MLL emphasizes calibration (accurate uncertainty estimates) in prediction over an accurate predictive mean. In Fig. 11, we show how the average validation set MLL changes with increasing training data. `SAL-HR` is the top-performing acquisition function on three out of six tasks, and rivals `BALD` for stability in predictive performance. This is particularly evident on the Ishigami function, where most methods fluctuate in the quality of their predictions. This can be attributed to emphasis on rapid hyperparameter learning, which is visualized in detail in App. D, Fig. 15. In the rightmost plot, the real-time average per-seed ranking of acquisition function performance is displayed as a function of the fraction of budget expended. `SAL-HR` performs best, followed by `BQBC` and `BALD`. `SAL-WS`, however, does not display similarly consistent predictive quality as `SAL-HR`. The ability of `SAL-HR` to correctly estimate hyperparameters ensures calibrated uncertainty estimates, which makes it the better candidate for BO. In App. C.1, Fig. 11, we show the evolution of the average Root Mean Squared Error (RMSE) of the same tasks, where `SAL-WS` performs best and `SAL-HR` lags behind, which demonstrates the viability of various distance metrics on different tasks.

## 4.2   Bayesian Optimization Tasks

For the BO tasks, we use the Hellinger distance for its proficiency in prediction calibration and hyperparameter learning. We compare against several state-of-the-art baselines from the BO literature: `NEI` for noisy experiments [36], as well as `JES` [29], the `MES` approach GIBBON [42] and PES [24]. As an additional reference, we include `NEI` for noisy experiments [36] using MAP estimation.

---

[1]`https://botorch.org/` (v0.8.4)

[2]All Normal and LogNormal distributions are parametrized by the mean and *variance*.

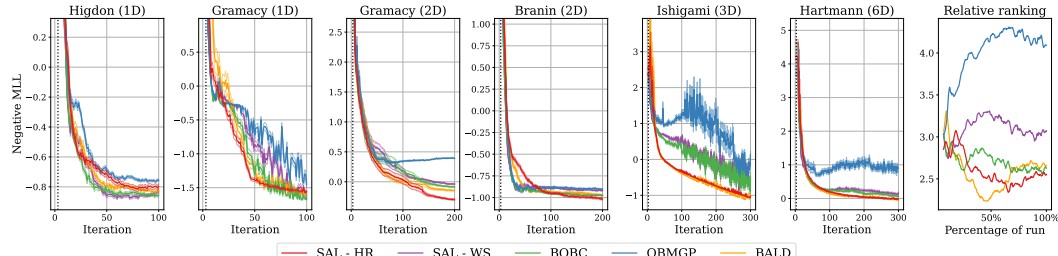

**Figure 4:** Negative Marginal Log Likelihood (MLL) on six active learning functions and the (smoothed) relative rankings throughout each run for `QBMGP`, `BQBC`, `BALD` and `SAL` using Wasserstein and Hellinger distance. We plot mean and one standard error for 25 repetitions.. `SAL-HR` is the top performing method, placing first in relative rankings. On Ishigami, only `SAL-HR` and `BALD` produces stable results.

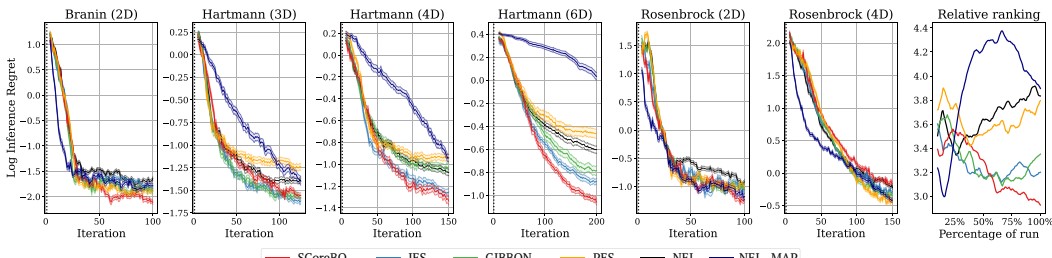

**Figure 6:** Average log inference regret and (smoothed) relative ranking across 50 repetitions between the acquisition functions for `SCoreBO`, `JES`, `MES` and `NEI` on six synthetic test functions. `SCoreBO` produces the best final regret on 4 out of 6 tasks, and has a substantially lower average ranking by the end of each run.

**Efficiently learning the hyperparameters**   To showcase `SCoreBO`'s ability to find the correct model hyperparameters, we run all relevant acquisition functions on samples from the 8-dimensional GP in Fig. 1. We exploit that for GP samples, the objectively true hyperparameters are known (in contrast to typical synthetic test functions). We utilize the same priors as in Fig. 1 on all the hyperparameters and compare `SCoreBO` to `NEI` to assess the ability of each acquisition function to work independently of the choice of prior. In Fig. 5, for each acquisition function, we plot the average log regret over 20 different 8-dimensional instances of this task. The tasks at hand have lengthscales that vary substantially between dimensions, as detailed in App B. The explanation for the good performance of `SCoreBO` can be see in Fig. 17 in App. D, where `SCoreBO` converges substantially faster towards the correct hyperparameter values than `NEI` for both types of priors.

**Synthetic test functions**   We run `SCoreBO` on a number of commonly used synthetic test functions for $25|\boldsymbol{\theta}|$ iterations, and present how the log inference regret evolves over the iterations in Fig. 6. All benchmarks are perturbed by Gaussian noise. We evaluate inference regret, i.e., the current best guess of the optimal location $\arg\max_{\boldsymbol{x}} \mu(\boldsymbol{x})$, which is conventional for non-myopic acquisition functions [22, 24, 29]. `SCoreBO` yields the the best final regret on four of the six tasks. In the relative rankings (rightmost plot), `SCoreBO` ranks poorly initially, but once hyperparameters are learned approximately halfway through the run, it substantially outperforms the competition. On Rosenbrock (4D), the relatively poor performance can explained by the appar-

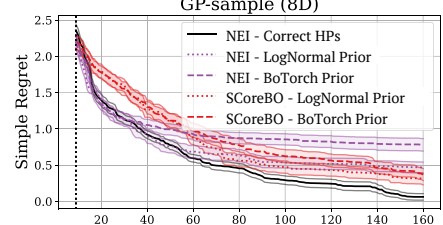

**Figure 5:** Regret for `NEI` and `SCoreBO` on the 8-dimensional GP sample for two different types of hyperparameter priors. Mean and standard deviation are plotted for all hyperparameter samples across 20 repetitions.

ent non-stationarity of the task, detailed in Fig. D.3, which makes hyperparameters diverge over time. This exposes a weakness of `SCoreBO`: When the modeling assumptions (such as stationarity) do not align with the task, optimization performance may suffer due to perpetual disagreement in the posterior. In App. C.2, we display the performance of `SCoreBO-KL` and `SCoreBO-WS` on the same set of benchmarks, where both display highly competitive performance.

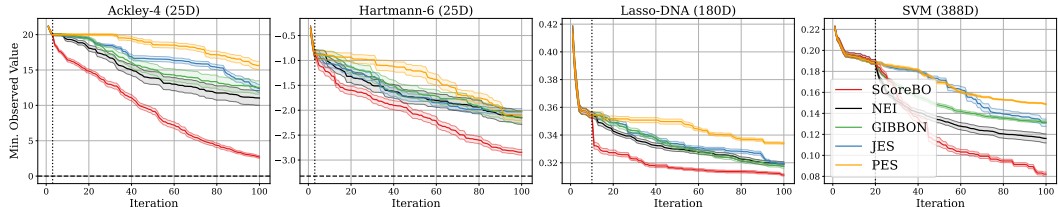

**Figure 7:** Final loss using SAASBO priors on the noisy embedded Ackley-4, embedded Hartmann-6, the DNA classification and the SVM HPO task, mean and one standard error. `SCoreBO` identifies the important dimensions rapidly, and successfully optimizes the tasks. The optimal value is marked with a dashed line.

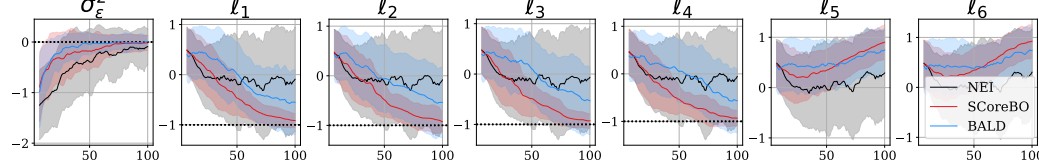

**Figure 8:** Hyperparameter convergence on the $25D$-embedded $4D$ Ackley function with a SAASBO HP prior for `SCoreBO`, `NEI` and `BALD`. Log HP mean and 1 standard deviation is plotted per iteration. `SCoreBO` identifies $\ell_1, \ldots, \ell_4$ as important (short lengthscales, $\ell_i \approx 10^{-1}$) with low uncertainty and $\ell_5, \ldots, \ell_{25}$ as dummy dimensions ($\ell_i \gg 10^1$). `NEI` fails to identify any important lengthscales, whereas `SCoreBO` correctly identifies active dimensions with high certainty.. Notably, `SCoreBO` finds accurate hyperparameters even faster than `BALD`, a pure active learning approach. Reference HP values (where available) are marked with a dashed line.

### 4.3 A Practical Need for Self-correction

Lastly, we evaluate the performance of `SCoreBO` on three atypical tasks with increased emphasis on the surrogate model: (1) high-dimensional BO through sparse adaptive axis-aligned priors (SAASBO) [15], (2) BO with additively decomposable structure (AddGPs) [19, 32] and (3) non-stationary, heteroskedastic modelling with HEBO [13]. Eriksson & Jankowiak [15] consider their proposed method for noiseless tasks, where active variables easily distinguish from their non-active counterparts. However, SAASBO is not restricted to noiseless tasks. For AddGPs, data cross-covariance, and lack thereof, is similarly difficult to infer in the presence of noise.

In Fig. 7, we visualize the performance of `SCoreBO` and competing acquisition functions *with SAASBO priors* on two noisy benchmarks, Ackley-4 and Hartmann-6, with dummy dimensions added, as well as two real-world benchmarks: fitting a weighted Lasso model in 180 dimensions [53], and the tuning of all 385 lengthscales and three regularization parameters of an SVM [12], a task also considered by Eriksson & Jankowiak [15]. On these benchmarks, where finding the correct hyperparameters is crucial for performance, `SCoreBO` clearly outperforms traditional methods. To further exemplify how `SCoreBO` identifies the relevant dimensions, in Fig. 8, we show how the hyperparameters evolve on the 25D-embedded Ackley (4D) task. `SCoreBO` quickly finds the correct lengthscales and outputscale with high certainty, whereas `NEI` remains uncertain of which dimensions are active throughout the optimization procedure. Impressively, `SCoreBO` finds accurate hyperparameters even faster than `BALD`, despite the latter being a pure active learning approach.

Secondly, we demonstrate the ability of `SCoreBO` to self-correct on *uncertainty in kernel design*, by considering AddGP tasks. We utilize the approach of Gardner et al. [19], where additive decompositions are marginalized over. Ideally, a sufficiently accurate decomposition is found quickly, which rapidly speeds up optimization through accurate cross-correlation of data. Fig. 9 demonstrates `SCoreBO`'s performance on two GP sample tasks and a real-world task estimating cosmological constants (leftmost 3 plots) and its ability to find the correct additive decompositions (right). We observe that `SCoreBO` identifies correct decompositions substantially better than `NEI`. Final performance, however, is only marginally better, as substantial resources are expended finding the right decompositions. Notably, the Cosmological Constants task does not display additive decomposability. As such, `SCoreBO` unsuccessfully expends resources attempting to reduce disagreement over additive structures, which hampers performance. This demonstrates that while `SCoreBO` learns the problem structure at increased rates, improved BO performance does not automatically follow.

Lastly, we apply `SCoreBO` to the `HEBO` [13] GP model, the winner of the NeurIPS 2020 Black-box optimization challenge [62]. The model employs input [56] and output warpings, the former of

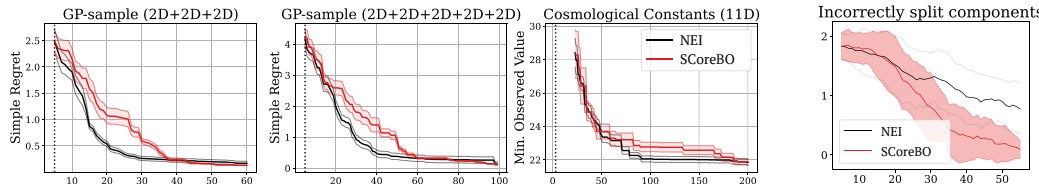

**Figure 9:** Final value of using AddGPs on 6D and 10D GP sample functions, fully decomposable in groups of two, and the Cosmological Constants tasks. SCoreBO achieves better final performance (left, middle) with low uncertainty, and successfully finds the additive components of the 6D task (right).

which are learnable to account for the heteroskedasticity that is prevalent in real-world optimization, and particularly HPO [13, 56], tasks. The complex model provides additional degrees of freedom in learning the objective. We evaluate `SCoreBO` and all baselines on three 4D deep learning HPO tasks: two involving large language models, and one from computer vision, from the PD1 [66] benchmarking suite. Fig. 10 displays that `SCoreBO` obtains the best final accuracy on 2 out of 3 tasks, suggesting that self-correction is warranted for optimization of deep learning pipelines.

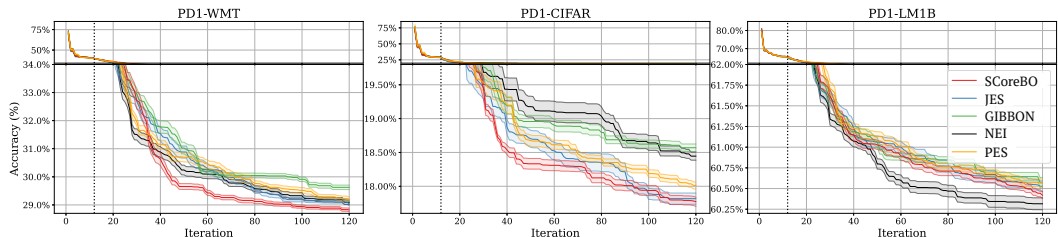

**Figure 10:** Performance on the PD1 deep learning tasks over 20 repetitions using the warpings from HEBO [13]. `SCoreBO` obtains the best final accuracy on 2 out of 3 tasks, placing second on the third.

## 5 Conclusion and Future Work

The hyperparameters of Gaussian processes play an integral role in the efficiency of both Bayesian optimization and active learning applications. In this paper, we propose Statistical distance-based Active Learning (`SAL`) and Self-Correcting Bayesian Optimization (`SCoreBO`), two acquisition functions that explicitly consider hyperparameter-induced disagreement in the posterior distribution when selecting which points to query. We achieve high-end performance on both active learning and Bayesian optimization tasks, and successfully learn hyperparameters and kernel designs at improved rates compared to conventional methods. `SCoreBO` breaks ground for new methods in the space of joint active learning and optimization of black-box functions, which allows it to excel in high-dimensional BO, where learning important dimensions are vital. Moreover, the potential downside of self-correction is displayed when the model structure does not support the task at hand, or when self-correction is not required to solve the task. For future work, we will explore additional domains in which `SAL` and `SCoreBO` can allow for increased model complexity in BO applications.

## 6 Limitations

`SCoreBO` displays the ability to increase optimization efficiency on complex tasks that necessitate accurate modeling. However, `SCoreBO`'s efficiency is ultimately contingent on the intrinsic ability of the GP to model the task at hand. Appendix 19 demonstrates this issue for the Rosenbrock (4D) function, where `SCoreBO` performs worse relative to other acquisition functions. There, the hyperparameter values increase over time instead of converge, which suggests that the objective is not part of the class of functions defined by the kernel. Thus, the self-correction effort is less helpful towards optimization. Moreover, increasing the model capacity, such as in Sec. 4.3, comes with increasing resources allocated towards self-correction. In highly constrained-budget applications, such resource allocation may not yield the best result, especially if increased model complexity is unwarranted. This is evident from the synthetic AddGP tasks, where despite accurately identifying the additive components, `SCoreBO` does not provide substantial performance gains over `NEI`. Lastly, `SCoreBO`'s reliance on fully Bayesian hyperparameter treatment makes it more computationally demanding than MAP-based alternatives, limiting its use in high-throughput applications.

## Acknowledgements

We thank the anonymous reviewers for their valuable contributions related SAL-KL and and its relationship with BALD, as well as their general feedback on the clarity of the paper and how our method was conveyed. We also thank Eytan Bakshy for their helpful feedback on earlier versions of this paper. Luigi Nardi was supported in part by affiliate members and other supporters of the Stanford DAWN project — Ant Financial, Facebook, Google, Intel, Microsoft, NEC, SAP, Teradata, and VMware. Carl Hvarfner, Erik Hellsten and Luigi Nardi were partially supported by the Wallenberg AI, Autonomous Systems and Software Program (WASP) funded by the Knut and Alice Wallenberg Foundation. Luigi Nardi was partially supported by the Wallenberg Launch Pad (WALP) grant Dnr 2021.0348. Frank Hutter acknowledges support through TAILOR, a project funded by the EU Horizon 2020 research and innovation programme under GA No 952215, by the Deutsche Forschungsgemeinschaft (DFG, German Research Foundation) under grant number 417962828, by the state of Baden-Württemberg through bwHPC and the German Research Foundation (DFG) through grant no INST 39/963-1 FUGG, and by the European Research Council (ERC) Consolidator Grant "Deep Learning 2.0" (grant no. 101045765). The computations were also enabled by resources provided by the Swedish National Infrastructure for Computing (SNIC) at LUNARC partially funded by the Swedish Research Council through grant agreement no. 2018-05973. Funded by the European Union. Views and opinions expressed are however those of the author(s) only and do not necessarily reflect those of the European Union or the ERC. Neither the European Union nor the ERC can be held responsible for them.



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

## A    Additional Related Work

We review approaches that address model accuracy in Bayesian optimization.

**Expanded Space of Models**    [38, 39] expands model uncertainty to kernel design in order to find the most accurate model possible within an expanded class of functions, whereas [56] expands model uncertainty to involve warpings of the input space. Since the expanded class of functions offers additional modelling flexibility, BO can ideally be conducted on a more accurate model than under a vanilla setting. As demonstrated with AddGPs in Sec. 4.3, SCoreBO can work in conjunction with kernel search to find such models at an accelerated rate.

**Simplified Modelling**    A contrasting line of work involve methods which restrict modeling by reducing the level of detail [6] or scope of the optimization [16, 44, 63]. This line of work acknowledges that modeling may not, or should not, be globally accurate or granular in order to conduct optimization efficiently within the allocated budget. As such, these lines of work address the issue of model accuracy by diametrically opposed philosophies. SCoreBO offers an orthogonal approach to model accuracy by *accelerating* the convergence of the model at hand, regardless of its level of complexity.

## B    Supplementary Material on Experiments

### B.1    Experimental Setup

All the relevant methods are implemented as acquisition functions in BoTorch. For both the active learning and BO experiments, we run NUTS [25] in Pyro [5] to draw samples from the GP posterior over hyperparameters. Tab. 3 displays the parameters of the MCMC in detail, as well as other relevant parameters of various MC estimations throughout the article. For the active learning experiments, we mimic the experimental setup used in Riis et al. [50], and put a log-normal distribution $\mathcal{LN}(0, 3)$ on the lengthscales, outputscale variance and noise variance. Furthermore, we consider the mean constant $c$ as a learnable parameter in the BO experiments, with a conventional $\mathcal{N}(0, 1)$ prior on the standardized inputs. When referring to the *BoTorch priors*, the priors are $\Gamma(3, 6)$, $\Gamma(2, 0.15)$, and $\Gamma(1.1, 0.05)$ for the lengthscales, outputscale, and noise variance, respectively, with the same prior on the learnable constant mean $c$.

**8D Gaussian Process sample task**    For the $8D$ GP sample task, we utilize the lognormal prior from Sec. B on the hyperparameters. The hyperparameters of the *sampled objective functions* are outlined in Tab. 1. As such, Tab. 1 displays the *true* hyperparameters of the task we are trying to optimize. These hyperparameters are referenced in Fig. 17 and Fig. 18

| Task | $\sigma^2$ | $\sigma_\varepsilon^2$ | $\ell$ | Kernel |
|---|---|---|---|---|
| GP sample - $8D$ | 1 | 0.1 | $\exp 10\{[-1, -0.5, -0.5, 0, 0, 0, 1.5, 1.5, 1.5]\}$ | Matérn |

**Table 1:** Hyperparameters of the $8D$ GP sample task.

**SAASBO experiments**    For the SAASBO experiments, we utilize Ax [3] , which runs the BoTorch [4] implementation of SAASBO with the deafult prior on the hyperparameters as per BoTorch version 0.8.4, which differs slightly from the paper by Eriksson & Jankowiak [15]. The lengthscale parameters are given a hierarchical prior as $\tau^2 \sim \mathcal{HC}(\alpha)$, $\kappa_i^2 \sim \mathcal{HC}(1)$ and $\ell_i = \frac{1}{\kappa_i \tau}$. We retain the default value of $\alpha = 0.1$. The noise variance, outputscale and mean constant are given the priors $\sigma_\varepsilon^2 \sim \Gamma(0.9, 10)$, $\sigma_f^2 \sim \Gamma(2, 0.15)$ and $c \sim \mathcal{N}(0, 1)$.

**Additive Gaussian Process experiments**    The Additive GP setup closely resembles that of [19]. An additive partitioning is sampled, and the marginal likelihood of the model is maximized with regard to $\boldsymbol{\theta} = \{\ell, \sigma_f, \sigma_\varepsilon\}$. We utilize a slightly adapted proposal distribution, and fix a maximal number of additive partitions $g_{max}$. Moreover, each dimension $d$ belongs to one distinct additive decomposition

---

[3]https://github.com/facebook/Ax
[4]https://github.com/pytorch/botorch

$g_d$, where $g_d \in [1, \ldots, g_{max}]$. At each iteration of the MCMC scheme, two dimensions $i, j = \{(i, j) \in [1, \ldots D], i \neq j\}$, are sampled uniformly at random and assigned the same new group index $g$, where $g \sim \mathcal{U}(1, g_{max})$. Setting $g_i = g, g_j = g$ thus proposes that dimensions $i$ and $j$ belong to the same additive decomposition. Under this proposal distribution, the proposed number of additive decompositions can never surpass $g_{max}$, but a lesser number of additive groups can be proposed. We utilize the same warm-starting mechanism as described in Gardner et al. [19], where at each iteration, the final accepted sample from the previous iteration acts as the initial proposal. The substitution of proposal distribution from the (unavailable) original implementation [19] was made to simplify a batch GP implementation in GPyTorch [20]. We choose to employ a BoTorch prior on the hyperparameters for all Additive GP tasks and use a squared exponential kernel, as described in [19].

| Task | $g_{max}$ |
|---|---|
| GP sample - $(2 + 2 + 2)D$ | 3 |
| GP sample - $(2 + 2 + 2 + 2 + 2)D$ | 5 |
| Cosmological constants | 4 |

**Table 2:** Additive tasks and their respective maximal number of maximal additive decompositions.

**HEBO Experiments** For the warped GP experiments, we adapt slightly more conservative priors due to the large number of hyperparameters of the model. Specifically, we set $\mathcal{LN}(0, 1)$ priors on the on lengthscales, outputscale variance and noise variance. For the input warpings, we employ a Kuwaraswamy distribution, a differentiable-CDF alternative to the input warpings proposed by Snoek et al. [56]. For each input dimension $j$, the untransformed input $x_j$ has a transformation applied as

$$z_j = (1 - x_j^{\alpha})^{\beta} \tag{11}$$

where $z_j$ is the resulting, transformed input. The warping parameters $\alpha$ and $\beta$ are both given a $\mathcal{LN}(0, 0.1)$ prior. We note that the HEBO experiments are susceptible to the number of hyperparameter sets. As such, we do not recommend running the experiments than fewer number of hyperparameter sets than that outlined by Tab. 3, as substantially fewer did not yield substantial empirical gains relative to vanilla BO.

| Task | Warmup | Thinning | No. hyperparameter sets | No. optima | No. RFFs |
|---|---|---|---|---|---|
| Active Learning | 256 | 16 | 16 | N/A | N/A |
| BO - Synthetic | 256 | 16 | 16 | 8 | 8192 |
| BO - GP samples | 256 | 16 | 16 | 8 | 2048 |
| BO - SAASBO | 128 | 8 | 16 | 8 | 8192 |
| BO - Additive GPs | 32 | 4 | 12 | 8 | 2048 |
| BO - Warped GPs | 64 | 6 | 32 | 8 | 2048 |

**Table 3:** MCMC hyperparameters for all experiments. For the AddGP experiments, each hyperparameter set involves the sampled additive decomposition *and* its associated MAP-trained hyperparameters. The total number of hyperparameter sets drawn are Warmup + Thinning * No. Hyperparameter sets.

### B.2 Benchmarks

For the active learning benchmarks, we follow Riis et al. [50] in the types of benchmarks and noise levels used. Each benchmark, as well as its search space, dimensionality and noise level is described in Tab. 4 and Tab. 5 for AL and BO, respectively. The noise level for all of the BO synthetic test functions were set to $\sigma_\varepsilon = 0.5$, except the Rosenbrock benchmarks, where the noise standard deviation was set to $\sigma_\varepsilon = 2.5$ due to the extremely large output range of the function. A smaller noise level would consequently bring the signal-to-noisy ratio under the permitted threshold supported by BoTorch.

**Compute resources.** All experiments are carried out on *Intel Xeon Gold 6130* CPUs. Each repetition of the tasks in Sec. 4.1 and Sec. 4.2 are run on 4 cores, and the tasks in Sec. 4.3 are run on 8 cores. Approximately $1,000$ core hours are used for each of the AL synthetic tasks, 2000 for the BO synthetic tasks, and 5000 for each task in Sec. 4.3.

| Task | Dimensionality | $\sigma_\epsilon$ | Search space |
|---|---|---|---|
| Gramacy | 1 | 0.1 | $[0.5, 2.5]$ |
| Higdon | 1 | 0.1 | $[0, 20]$ |
| Gramacy | 2 | 0.05 | $[-2, 6]^D$ |
| Branin | 2 | 11.32 | $[-5, 10] \times [0, 15]$ |
| Ishigami | 3 | 0.187 | $[-\pi, \pi]^D$ |
| Hartmann-6 | 6 | 0.0192 | $[0, 1]^D$ |

**Table 4:** Benchmarks used for the active learning experiments.

| Task | Dimensionality | $\sigma_\epsilon$ | Search space |
|---|---|---|---|
| Branin | 2 | 0.5 | $[-5, 10] \times [0, 15]$ |
| Rosenbrock-2 | 2 | 2.5 | $[-1.5, 1.5]^D$ |
| Hartmann-3 | 6 | 0.5 | $[0, 1]^D$ |
| Rosenbrock-4 | 4 | 2.5 | $[-1.5, 1.5]^D$ |
| Hartmann-4 | 4 | 0.5 | $[0, 1]^D$ |
| Hartmann-6 | 6 | 0.5 | $[0, 1]^D$ |

**Table 5:** Benchmarks used for the Bayesian optimization experiments.

## C    Additional Experiments

We display the RMSE performance of each of the `SAL` variants, a comparison of the MC and MM variants of `SAL` and `SCoreBO`.

### C.1    AL RMSE Performance

Fig. 11 displays the performance of both `SAL` variants and benchmark AL acquisition functions for the same set of tasks as in Sec. 4.1. We observe that `SAL-WS` consistently displays top performance, whereas `SAL-HR` lags behind substantially. This showcases  `SAL-HR`'s emphasis on hyperparameter learning as opposed to global exploration. By accurately assessing hyperparameters while sacrificing global exploration,  `SAL-HR` ensures predictions with *calibrated uncertainty*, while sacrificing the *accuracy in predictive mean* that follows from exploring the search space.   `SAL-WS` offers a compromise which performs well under both metrics, sacrificing hyperparameter learning and calibration for accuracy in predictive mean.

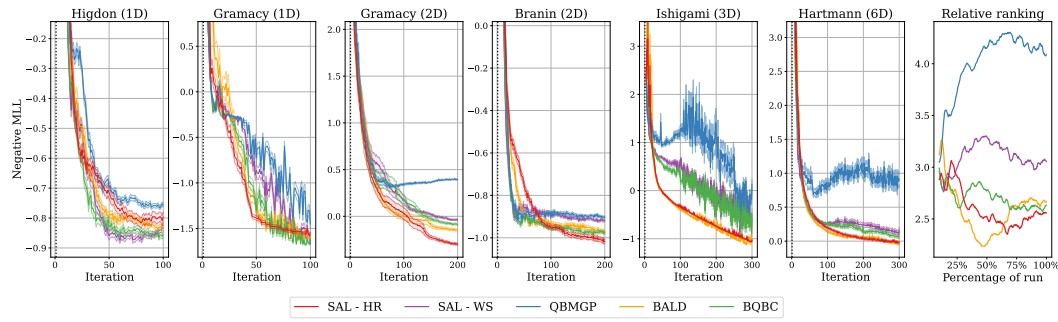

**Figure 11:** Root Mean Squared Error (RMSE) on six active learning functions and the (smoothed) relative rankings throughout each run for `QBMGP`, `BQBC`, `BALD` and `SAL` using Wasserstein and Hellinger distance. We plot mean and one standard error for 25 repetitions..  `SAL-WS` is the top performing method, placing first in relative rankings. On Ishigami, only  `SAL-HR` and `BALD` produces stable results.

### C.2    SCoreBO Distance Measure Ablation Analysis

In Fig. 13, We compare the Hellinger and Wasserstein variants of `SCoreBO`, both utilizing the MC approximation of the statistical distance. Since the MC approximation is asymptotically exact, we can better assess the performance of each distance metric, without having to consider the confounding factor that the MM approximation introduces. We note that `SCoreBO-WS` outperforms `SCoreBO-HR`

on two tasks, but `SCoreBO`-HR is the overall more consistent approach. We hypothesize that the relative failure of `SCoreBO`-WS on Rosenbrock (4D) is caused by the objective's non-stationarity, which likely causes exceedingly large exploration of the hyperparameter space. This is supported by Fig. D.3, where the hyperparameters diverge over time on the Rosenbrock function.

Furthermore, in Fig. 12, we compare the KL and Hellinger variants of `SCoreBO`, both utilizing the moment matching estimation of the posterior. Notably, `SCoreBO`-KL is the natural extension of `BALD` to the self-correcting framework, in line with Prop. 1. `SCoreBO`-HR performs marginally better than `SCoreBO`-KL, winning 4 out of 6 tasks. However, the two variants are relatively close.

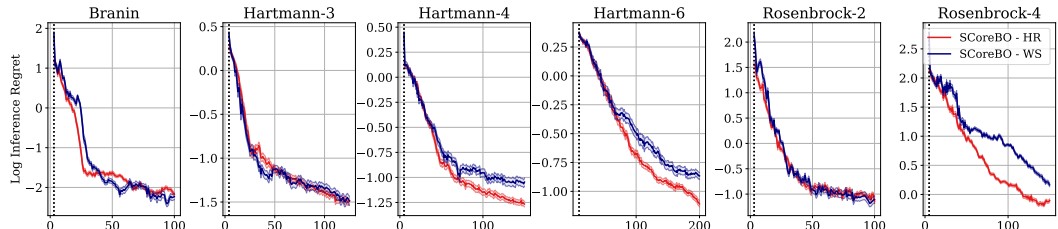

**Figure 12:** Log Regret of the Hellinger and Wasserstein MC-variants of `SCoreBO`. Both variants are competitive on all benchmarks, except for Wasserstein on Rosenbrock (4D) which lags behind slightly. Overall, Hellinger is more constistent, and wins 4 out of 6 benchmarks.

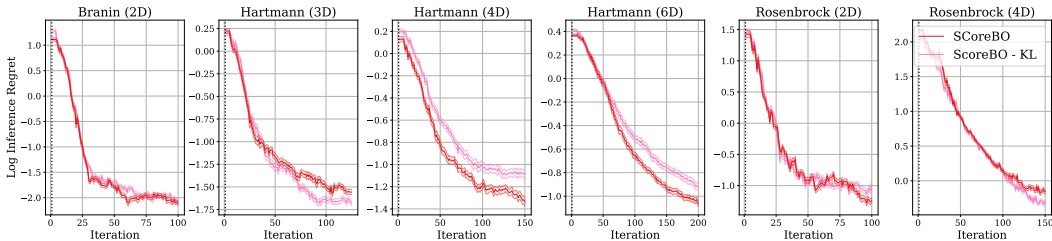

**Figure 13:** Log Regret of the Hellinger and KL moment matching variants of `SCoreBO`. Both variants are competitive on all benchmarks. Overall, Hellinger is marginally better, and wins 3 out of 6 benchmarks with an approximate tie on Branin.

### C.3 SAASBO Noise Ablations

In Fig. 13, We measure the performance of `SCoreBO` and `NEI` on the embedded 25-dimensional Ackley (4D) and Hartmann (6D) tasks with varying noise levels using the SAAS [15] prior. We run three noise levels for each task: Noiseless (solid line), low (dashed line), where the noise standard deviation corresponds to 3% of total output range for Hartmann (6D), and 1.3% for Ackley, and high (dotted line, 13.3% / 4%). With increasing levels of noise, the difficulty of inferring active dimensions is expected to increase substantially, which should in turn hamper BO performance. In Fig. 14, we see that `NEI` and `SCoreBO` perform comparably on noiseless tasks (solid line) finding close-to-optimal solutions within 100 iterations. Moreover, for small levels of noise), the performance is still comparable. However, we observe a drastic fall-off in performance for `NEI` at the highest noise level, whereas `SCoreBO`'s degrades gracefully. Notably, `SCoreBO` almost retains the performance of the noiseless at the highest noise level for Ackley.

## D   Hyperparameter convergence

In Figures 15, 16, 17, 18, and 19 We demonstrate examples of hyperparameter convergence in AL and BO, as well as an example of hyperparameter divergence on the Rosenbrock ($4D$) function, where `SCoreBO` performs marginally worse.

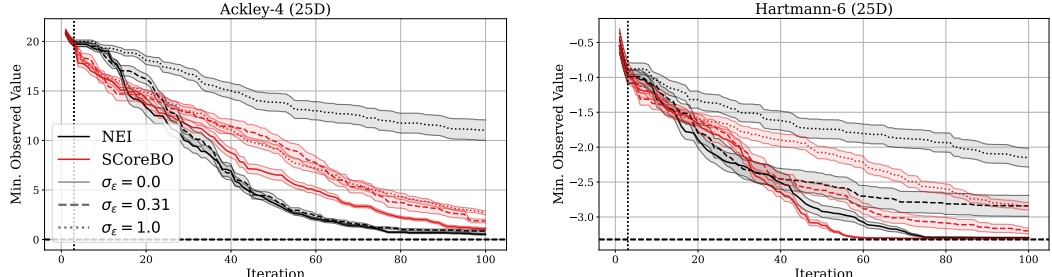

**Figure 14:** Best observed value for varying levels of observation noise for `NEI` and `SCoreBO` using SAAS priors on the 25D-embedded lower-dimensional test functions. For Low levels of noise, the performance of `NEI` and `SCoreBO` are comparable, but `SCoreBO` retains performance substantially better for higher levels for noise.

## D.1 Active Learning Tasks

We display the hyperparameter convergence of `SAL-WS`, `SAL-HR` and the baseline active learning acquisition functions in Fig. 15. Both variants display accelerated hyperparameter learning compared to `BQBC`. `SAL-HR` in particular achieves low-variance hyperparameter uncertainty on Ishigami and the higher-dimensional Hartmann-6, where other methods struggle. We obtain approximately correct hyperparameters for these tasks by randomly sampling 300 points on the noiseless benchmark, thereafter performing MCMC and averaging the sampled hyperparameter estimates in logspace. The noise level is known a priori. and estimates the other hyperparameters with substantially greater certainty than other methods. We note that there is a drift in the hyperparameters as the number of observations increase, where output- and lengthscales trade off to reduce model complexity. As such, we provide an approximately stationary alternative in Fig. 16, where the outputscale is removed to avoid drift. In both cases, `SAL-HR` displays superior hyperparameter convergence, obtaining accurate hyperparameters in far fewer iterations, and with substantially less uncertainty than the alternatives.

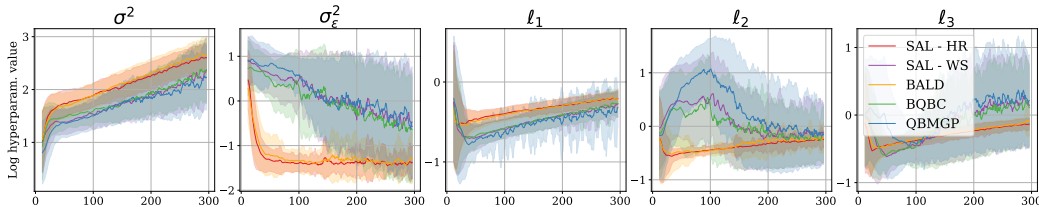

**Figure 15:** Hyperparameter convergence on the Ishigami test function with outputscale. While no acquisition converges, `SAL-HR` and `BALD` display substantially more stable hyperparameters than other approaches.

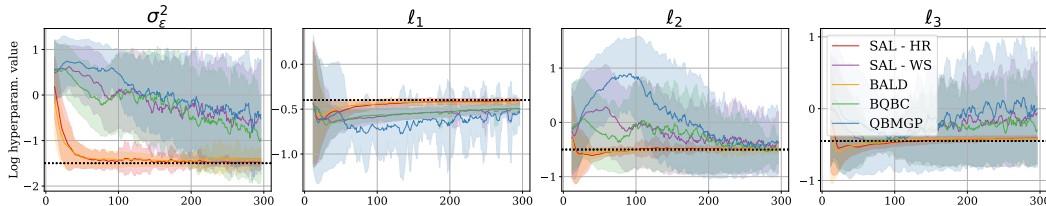

**Figure 16:** Hyperparameter convergence on the Ishigami test function without outputscale. `SAL-HR` and `BALD` display stable hyperparameter convergence, and are the only acquisition function to accurately estimate all parameters.

## D.2 GP sample tasks

We display the convergence of `SCoreBO` and `NEI` with a wide lognormal and BoTorch prior on the GP sample task. We observe that the lognormal prior is well-aligned for most hyperparameters, whereas BoTorch prior is misaligned. This is evidenced by the unimportant dimensions $\ell_6, \ell_7,$ and $\ell_8$, which have suggested lengthscales that are incorrect by more than an order of magnitude. Nevertheless, `SCoreBO` suggests lengthscales that are approximately twice as long $(10^{0.25} \approx 1.8)$ as `NEI`$(10^{-0.05} \approx$

0.9), and thus avoids unneessary exploration along these dimensions. Moreover, SCoreBO correctly identifies the most important dimensions $\ell_1, \ell_2$, and $\ell_3$ with good accuracy quickly, whereas NEI struggles to identify $\ell_1$. SCoreBO slightly overestimates the importance of dimensions 2 and 3, likely to compensate for the inability to accurately estimate the importance of other hyperparameters.

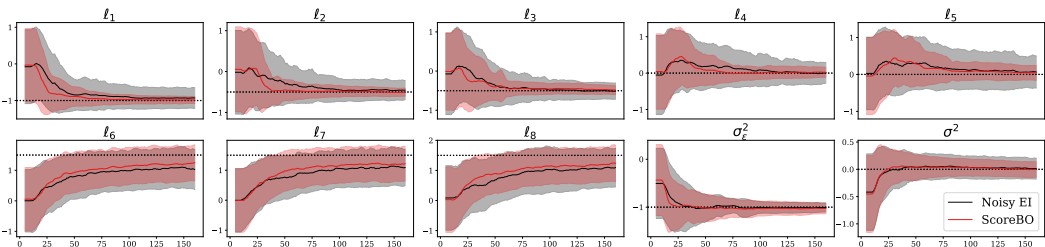

**Figure 17:** Hyperparameter convergence on the 8-dimensional GP sample for the broad log-normal prior. The black dashed line indicates true hyperparameter values. Mean and standard deviation are plotted across 20 repetitions, and a 3 iteration moving average of the plotted moments is applied to increase readability. Lengthscales $\ell_d$ ordered smallest (most important) to largest (least important). SCoreBO finds accurate hyperparameters faster, has the most accurate values for all hyperparameters, and has substantially lower variance for all important (i.e. not $\ell_6, \ell_7$, and $\ell_8$) hyperparameters except for the noise variance.

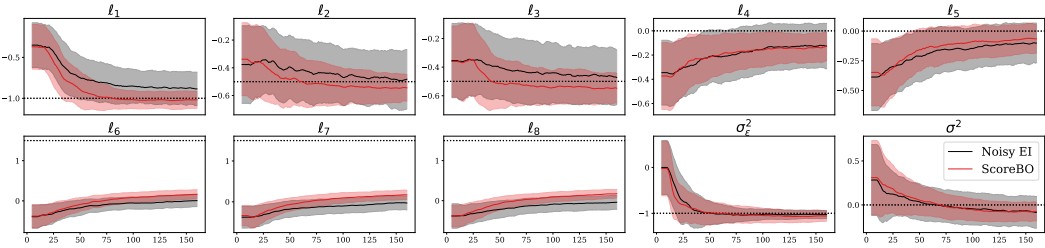

**Figure 18:** Hyperparameter convergence on the 8-dimensional GP sample for the BoTorch priors. The black dashed line indicates true hyperparameter values. Mean and standard deviation are plotted across 20 repetitions, and a 3 iteration moving average of the plotted moments is applied to increase readability. Lengthscales $\ell_d$ ordered smallest (most important) to largest (least important). SCoreBO finds accurate hyperparameters faster, has the most accurate values for all hyperparameters, and has substantially lower variance for all important (i.e. not $\ell_6, \ell_7$, and $\ell_8$) hyperparameters except for the noise variance.

### D.3 Hyperparameter Divergence on Synthetic BO tasks

We highlight additional examples on synthetic BO test functions where hyperparameters diverge. Due to the non-stationary structure of Rosenbrock in particular (and to a lesser extent, Branin), hyperparameters values diverge as the number of observations increase. In particular, the extreme steepness along the edges suggests an exceedingly large outputscale. With increasing observations, a lengthscale-outputscale trade-off occurs, where both hyperparameters grow seemingly indefinitely. Notably, this behavior is consistent regardless of the acquisition function (BO, AL, SOBOL). Due to the restricted hyperparameter set employed in the AL tasks, this problem is distinct to the BO tasks.

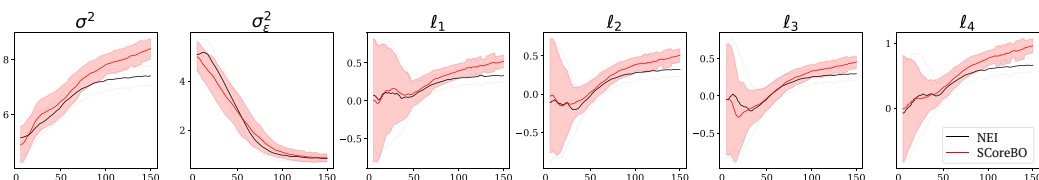

**Figure 19:** Hyperparameter divergence for SCoreBO and NEI on Rosenbrock (4D). The outputscale grows larger with increasing iterations, and the lengthscales grow similarly large as a countermeasure.

# E Approximation Strategies

We display the quality of the moment matching approximation for both the Hellinger and Wasserstein distance. Moreover, we compare the performances of the MM and MC approaches.

## E.1 MC approximation of `SAL` and `SCoreBO`

Using Monte Carlo, different distances are most efficiently estimated in different manners. To approximate the Wasserstein distance, we utilize quasi-Monte Carlo. From the definition of the distance in one dimension, we obtain

$$W^2(p,q) = \int_0^1 |Q(u) - P(u)|^2 du \approx \sum_{\ell=1}^L |Q(u_\ell) - P(u_\ell)|^2, \tag{12}$$

where $u_\ell \sim \mathcal{U}(0,1)$, and $P(x)$ and $Q(x)$ are the respective cumulative distributions for $p(x)$ and $q(x)$. To approximate the Hellinger distance, we obtain

$$H^2(p,q) = 1 - \int_{\mathcal{X}} \sqrt{\frac{q(x)}{p(x)}} p(x) dx \approx 1 - \sum_{\ell=1}^L \sqrt{\frac{q(x_\ell)}{p(x_\ell)}}, \tag{13}$$

where $x_\ell \sim p(x)$ is sampled using MC. In `SCoreBO`, $p(x)$ is the marginal $p(y_{\boldsymbol{x}}|\mathcal{D})$, and $q(x)$ each of the various conditionals $p(y_{\boldsymbol{x}}|*, \boldsymbol{\theta}, \mathcal{D})$.

## E.2 Performance of Monte Carlo

We display the performance of the MC variants of `SAL`-WS and `SCoreBO`-HR compared to their MM counterparts. Overall, performances are comparable, as each variant slightly exceeds the other on a couple of benchmarks. On the most complex benchmarks (Ishigami, Hartmann-4, Hartmann-6), the MC variant outperforms MM slightly, which suggests that MC is increasingly justified as disagreement in the posterior gets larger.

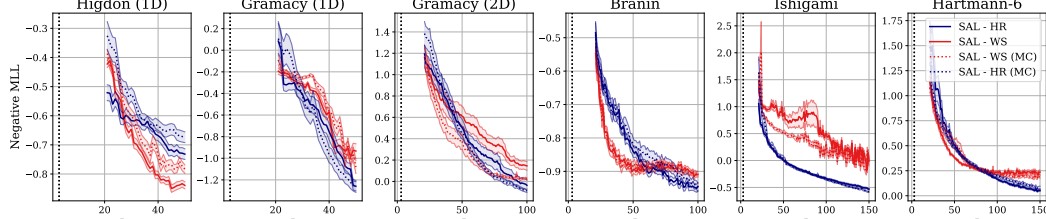

**Figure 20:** Negative marginal log likelihood (MLL) of the SAL MC (blue) and MM (red) variants on the active learning benchmarks. Overall performance is comparable, with three effectively tied benchmarks. MC outperforms slightly Hartmann-4 and Hartmann-6, and MM on Hartmann-3.

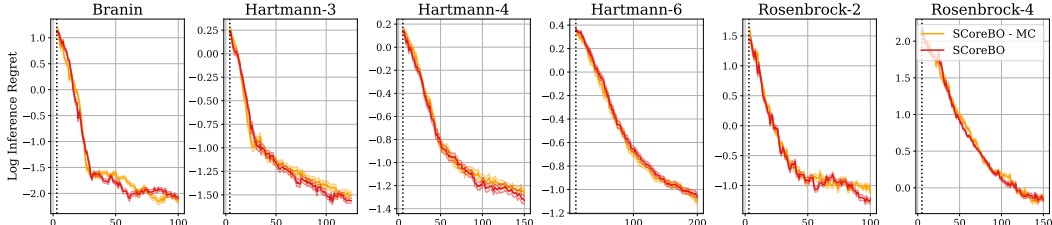

**Figure 21:** Log regret of `SCoreBO`-MM and `SCoreBO`-MC on the synthetic BO benchmarks. Overall performance is comparable, with MM outperforming marginally on 4 out of 6 tasks. MC notably outperforms slightly on the difficult Ishigami test function.

## E.3 Hellinger Distance Approximation

We display the accuracy of the moment matching approximation, and the sensitivity of the MC approximation to the number of samples $L$. In Fig. 24 and Fig. 25, we highlight two examples of the moment matching approximation in comparison to a large-scale, asymptotically exact variant of the MC approximation with 2048 samples. In Fig. 24, the MM approximation struggles to capture the sharp, multimodal surfaces in (blue), and consistently overestimates the distance in (orange). In Fig. 25, the included conditional posteriors are substantially more similar, and as such, the moment matching approximation is more accurate. The shape of the acquisition function is captured almost perfectly, and the magnitude is only marginally overestimated, most prominently in (green). We display the

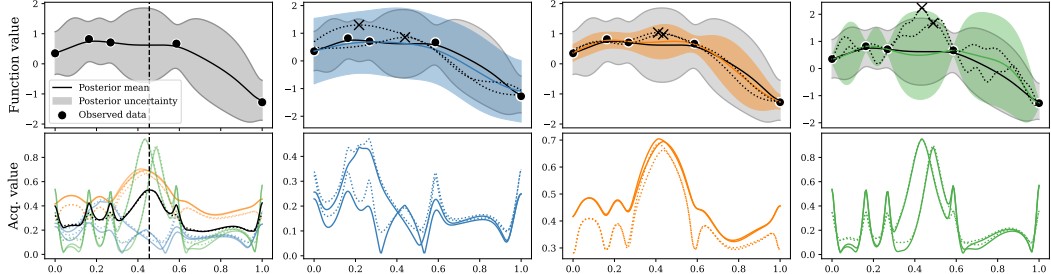

**Figure 22:** Example of the per-sample Hellinger distance computation using moment matching (solid lines) and large-scale, asymptotically exact quasi-MC with 2048 samples. The moment matching approximation mostly retains the shape of the asymptotically exact variant. However, it does not perfectly capture the multi.modality in (blue), and overestimates the distance in the low-variance region at the right edge of (orange). The acquisition function y-axis is scaled individually per model to better highlight the difference in acquisition function value.

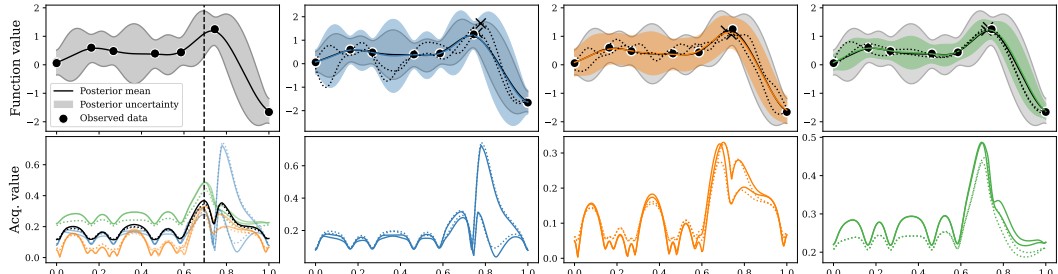

**Figure 23:** Example of the per-sample Hellinger distance computation using moment matching (solid lines) and large-scale, asymptotically exact quasi-MC with 2048 samples. The moment matching approximation captures the shape of the asymptotically exact variant well, but overestimates the distance slightly in (green). The acquisition function y-axis is scaled individually per model to better highlight the difference in acquisition function value.

## E.4 Wasserstein Distance Approximation

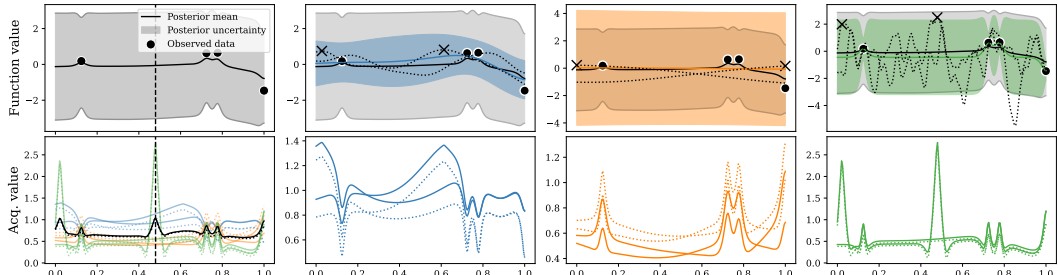

**Figure 24:** Example of the per-sample Wasserstein distance computation using moment matching (solid lines) and large-scale, asymptotically exact quasi-MC with 2048 samples. The moment matching approximation mostly retains the shape of the asymptotically exact variant. The shape of the acquisition function is generally well captured, but high-variance regions have their distance underestimated by the moment matching approach, and low-variance regions have their distance over-estimated, leading to a biased approximation. The acquisition function y-axis is scaled individually per model to better highlight the difference in acquisition function value.

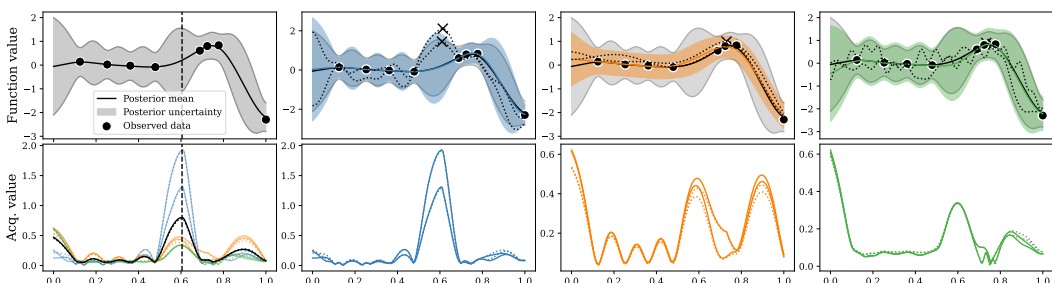

**Figure 25:** Example of the per-sample Wasserstein distance computation using moment matching (solid lines) and large-scale, asymptotically exact quasi-MC with 2048 samples. The moment matching approximation captures the shape of the asymptotically exact variant well, and only marginally over- and underestimates the distance. The acquisition function y-axis is scaled individually per model to better highlight the difference in acquisition function value.

# F   Equivalence to BALD

We show that SAL, using the KL divergence as the metric $d$, is equivalent to BALD.

$$
\begin{aligned}
BALD = I(y; \boldsymbol{\theta}) &= H(p(y)) - E_{\boldsymbol{\theta}}[H(p(y|\boldsymbol{\theta}))] = \\
&= -\int_{-\infty}^{\infty}\int_{\boldsymbol{\theta}} p(\boldsymbol{\theta})p(y|\boldsymbol{\theta})\, log[p(y)]d\boldsymbol{\theta}dy + \int_{\boldsymbol{\theta}} p(\boldsymbol{\theta})\int_{-\infty}^{\infty} p(y|\boldsymbol{\theta})log[p(y|\boldsymbol{\theta})]dyd\boldsymbol{\theta} = \\
&= \int_{\boldsymbol{\theta}} p(\boldsymbol{\theta})\int_{-\infty}^{\infty} p(y|\boldsymbol{\theta})log\left[\frac{p(y|\boldsymbol{\theta})}{p(y)}\right]dyd\boldsymbol{\theta} = \int_{\boldsymbol{\theta}} p(\boldsymbol{\theta})KL(p(y|\boldsymbol{\theta})||p(y))d\boldsymbol{\theta} = \\
&= E_{\boldsymbol{\theta}}[KL(p(y|\boldsymbol{\theta})||p(y))] = SAL.KL
\end{aligned}
$$

# G   Additional Background

We provide additional background on fully Bayesian hyperparameter treatment in Gaussian processes and details regarding statistical distance metrics.

## G.1   Fully Bayesian Treatment

The posterior probability of observing a value $y_{\boldsymbol{x}}$ for a point $\boldsymbol{x}$ is given as:

$$p(y_{\boldsymbol{x}}|\mathcal{D}) = \int_{\boldsymbol{\theta}} \int_f p(y_{\boldsymbol{x}}|f,\boldsymbol{\theta})p(f|\boldsymbol{\theta},\mathcal{D})p(\boldsymbol{\theta}|\mathcal{D})df\,d\boldsymbol{\theta}$$

$$= \int_{\boldsymbol{\theta}} \int_f p(y_{\boldsymbol{x}}|f,\boldsymbol{\theta})p(f|\boldsymbol{\theta},\boldsymbol{x},\mathcal{D})p(\boldsymbol{\theta}|\mathcal{D})df)d\boldsymbol{\theta},$$

where $f$ are the noiseless, latent function values as $\boldsymbol{x}$ and $\mathcal{D}$ is the observed data. The inner integral is equal to the GP predictive posterior,

$$\int_f p(y_{\boldsymbol{x}}|f,\boldsymbol{\theta})p(f|\boldsymbol{\theta},\mathcal{D})df = p(y_{\boldsymbol{x}}|\mathcal{D},\boldsymbol{\theta}).$$

However, the outer integral is intractable and is estimated using Markov Chain Monte Carlo (MCMC) methods. The resulting posterior prediction

$$p(y_{\boldsymbol{x}}|\mathcal{D}) = \int_{\boldsymbol{\theta}} p(y_{\boldsymbol{x}}|\mathcal{D},\boldsymbol{\theta})p(\boldsymbol{\theta}|\mathcal{D})d\boldsymbol{\theta} \approx \frac{1}{M}\sum_{j=1}^M p(y_{\boldsymbol{x}}|\mathcal{D},\boldsymbol{\theta}_j), \ \ \boldsymbol{\theta}_j \sim p(\boldsymbol{\theta}|\mathcal{D}),$$

is a Gaussian Mixture Model (GMM).

Within BAL and BO, the fully Bayesian treatment is often extended to involve the acquisition function, such that the acquisition function $\alpha$ is computed as an expectation over the hyperparameters [46, 55]

$$\alpha(\boldsymbol{x}|\mathcal{D}) = \mathbb{E}_{\boldsymbol{\theta}}[\alpha(\boldsymbol{x}|\boldsymbol{\theta},\mathcal{D})] \approx \frac{1}{M}\sum_{j=1}^M \alpha(\boldsymbol{x}|\boldsymbol{\theta}_j,\mathcal{D}) \ \ \boldsymbol{\theta}_j \sim p(\boldsymbol{\theta}|\mathcal{D}).$$

This is also the definition of fully Bayesian treatment considered in this work.

### G.2  Statistical Distance Details

**The Hellinger distance**   is a similarity measure between two probability distributions which has previously been employed in the context of BO-driven automated model selection by Malkomes et al. [39]. For two probability distributions $p$ and $q$, it is defined as

$$H^2(p,q) = \frac{1}{2}\int_{\mathcal{X}} \left(\sqrt{p(x)} - \sqrt{q(x)}\right)^2 \lambda dx, \tag{14}$$

with some auxiliary measure $\lambda$ with which both $p$ and $q$ are absolutely continuous. Specifically, for two normally distributed variables $z_1 \sim \mathcal{N}(\mu_1,\sigma_1^2), \ z_2 \sim \mathcal{N}(\mu_2,\sigma_2^2)$,

$$H^2(z_1,z_2) = 1 - \sqrt{\frac{2\sigma_1\sigma_2}{\sigma_1^2+\sigma_2^2}}\exp\left[-\frac{1}{4}\frac{(\mu_1-\mu_2)^2}{\sigma_1^2+\sigma_2^2}\right]. \tag{15}$$

The Hellinger distance seeks to minimize the ratio between difference in mean and the sum of variances, which punishes outlier predictive distributions of high confidence. Similar to KL, initial queries have a tendency to be axis-aligned to attain selective length scale information.

**The Wasserstein distance**   is the average distance needed to move the probability mass of one distribution to morph into the other. The Wasserstein-$k$ distance is defined as

$$W_k(p,q) = \left(\int_0^1 |F_q(x) - F_p(x)|^k dx\right)^{1/k} \tag{16}$$

For the normal distributions $z_1$ and $z_2$, the Wasserstein-2 distance is defined as

$$W_2(z_1,z_2) = \sqrt{(\mu_1-\mu_2)^2 + (\sigma_1-\sigma_2)^2}. \tag{17}$$

In practice, $W_2$ places a premium on matching large-variance regions, leading to higher global exploration which can be detrimental for global optimization.

**The KL divergence** The KL divergence is a standard asymmetrical measure for dissimilarity between probability distributions. For two probability distributions $P$ and $Q$, it is given by $\mathcal{D}_{KL}(P \parallel Q) = \int_{\mathcal{X}} P(x)\log(P(x)/Q(x))dx$. For Gaussian variables, it is computed as

$$KL(z_1 \| z_2) = \log\frac{\sigma_1}{\sigma_1} + \frac{\sigma_1^2 + (\mu_1 - \mu_1)^2}{\sigma_1^2} - \frac{1}{2} \tag{18}$$

The KL divergence mainly prioritizes same order-of-magnitude variances, and will initially query the same location multiple times to assess noise levels. Thereafter, it tends to query in an axis-aligned fashion, close to previous queries, to attain information regarding the length scales, but places a low priority on global exploration.

