# OpenReview forum: "Self-Correcting Bayesian Optimization through Bayesian Active Learning"
_NeurIPS.cc/2023/Conference — NeurIPS 2023 poster_

### Official Review · Reviewer_BCnP · 2023-06-18

**Soundness:** 3 good
**Presentation:** 2 fair
**Contribution:** 2 fair
**Rating:** 5
**Confidence:** 3

**Summary:**

The paper proposed a novel function SAL used for active learning based on distribution difference. Then the author further incorporated the SAL function into Bayesian optimization (BO) to improve BO performance. Most numerical experiments are conducted on synehtic benchmakr functions. Better performances are achieved by using the proposed method compared to several baseline functions.

**Strengths:**

Overall, the paper is easy to follow despite several confusions.

**Weaknesses:**

Lack of real-world experiments.

I would suggest the authors be precisely clear on their symbols.

I have some questions about the technical details.

**Questions:**

In general, I grasp the main idea of this paper. However, I suggest that the authors distinguish among three different things: (i) the true underlying function $f(x)$, (ii) the observed noisy function value $f(x)+\epsilon$, and (iii) the GP model $GP(x)$, which is an approximation to $f(x)$. The authors should carefully examine the usage of each symbol throughout their paper and ensure that they properly distinguish between (i)-(iii). Currently, the authors sometimes misuse these symbols, which hinders the understanding of the paper.

Below, I have listed several questions and would appreciate it if the authors could provide answers. This will help me gain a better understanding and assessment of the paper.

1. The authors state that the mean function $m(\cdot)$ is set to a constant. Why is this constant not included in the set of hyperparameters?

2. The numerical experiments primarily focus on benchmark functions, with the exception of DNA classification task and cosmological estimation. It would be beneficial to include more real-world experiments. For example, Bayesian optimization is commonly used for tuning network parameters in automl. It would be valuable to add experiments similar to that.

3. What is the specific form of $p(\theta|D)$ in step 4 shown in Algorithm 1?

4. What is $p(f|\theta,D)$ in Step 6? And what does Step 6 do? The symbol $f$ is used to represent the underlying function in equation (1). I am confused about how to sample an $f$ from $p(\cdot|\theta,D)$. Since $f(\cdot)$ represents the underlying function and does not depend on the GP model, I assume that $p(f|\theta,D)$ means sampling a GP instance given $\theta$ and $D$. If this is the case, I suggest the authors use another symbol instead of $f$ here.

5. One advantage of Bayesian optimization is that its acquisition function does not require calling the real function. Instead, it only needs predictions based on the GP (e.g., EI = mean of GP - std of GP). Thus, Bayesian optimization can save functional evaluation times. Typically, Bayesian optimization is employed in situations where observing the function value $y_x$ at a specific $x$ is time-consuming or expensive. However, I am confused here. Does $y_x$ represent the observed function value at $x$ according to equation (1)? Does this mean that in order to evaluate equation (9), we actually need to simulate or obtain $y_x$ at a specific $x$? In other words, does the proposed acquisition function (9) require the observation of $y_x$? If so, it seems to me that the method is not valid, as it sacrifices the fundamental benefit of Bayesian optimization.

6. Continuing from the previous question, how do we evaluate the distribution value $p(y_x|\cdots)$ in Step 7? Moreover, how do we evaluate $p(y_x|D)$ in (9)?

7. What is the difference between $p(y_x|\theta,D)$ used in equation (7) and $p(f|\theta,D)$ used in Step 6? Are they different in terms of considering $\epsilon$ or not (i.e., in equation (1), $y_x=f(x)+\epsilon$)?

8. I am confused about the condition on $(x^\star,f^\star)$. What do the authors mean by $(x^\star,f^\star)$? Are they the true optimals related to the underlying function $f$ or are they the optimals related to a GP model?

9. The considered numerical experiments all seem rather low-dimensional.

10. I am concerned about the insufficient technical contribution. It appears to me the main contribution could be summarized in one equation, Eq. (7). Previous similar papers usually also provided theoretical guarantees (e.g., the JES paper).  Solely a new function form seems rather insufficient as a standalone paper. I would appreciate it if the author could justify their contribution.

**Writing suggestions**

1. I assume "HP" in Figure 1 means "hyper-parameter." However, at first sight, "HP" is confusing, especially since many words have abbreviations as "HP." I suggest that the authors write the full name when it first appears.

2. Line 72 contains a typo: "$\epsilon\sim N(0,\sigma^2_\epsilon)$" instead of "$\epsilon^2$."

3. Keep the symbol format consistent: In line 76, $\theta$ is not bold, while in line 80, $\theta$ is bold.

4. The Hellinger distance has the general form shown in equation (4). Why is there no general form provided for the Wasserstein distance?

5. In line 221, "robust. than" needs to be revised for clarity.

**Limitations:**

The authors provided limitations in the last section.

---

> ### Author Rebuttal · Authors · 2023-08-08
>
> We thank the reviewer for their thorough work reviewing the paper. We have added additional experiments, and hope to clarify the notation-induced confusion.
>
>
> __0)__ _Distinguish among three different things: $f(x), f(x) + ε, GP(x)$_
>
>
> Using the notation $f(x)$ to denote both the true black-box function _and_ a random variable modeled by the GP can be confusing. __This is, however, standard practice in BO literature__ [Hernandez-Lobato et.al, 2014, Wang and Jegelka, 2017, Hvarfner et. al. 2022, Takeno et. al. 2022] and we are hesitant to break this convention.
>
> To clarify:
> - $p(f |D)$ is the distribution over functions that the GP induces, and a sample draw from this distribution is a function $f_i \sim p(f |D)$ (as in Alg. 1 step 6).
> - $y_x$ (reads $y$ at $x$) is a random variable representing the noisy output at $x, y_x = f (x) + \epsilon$. Its __predictive posterior under the GP__ is $p(y_x|D)$.
>
> Hopefully, this clarifies our notation to the reviewer.
>
> __1)__
> This is well spotted by the reviewer. We do indeed marginalize over the mean (Appendix A), but we should have declared it as a HP in Section 2.1. This is part of the HPs that SAL and SCoreBO learn. We will update the CR.
>
> __2)__
> We have added __three (4D) HPO tuning tasks__ from the PD1 benchmarking suite — Two involving large language models, and one from computer vision. We adopt the same setting as in HEBO [Cowen-Rivers et. al. 2020], where input and output warpings [Snoek et, al 2014] are used to account for the heteroskedasticity prevalent in real HPO tasks. We report the results in the PDF material submitted with this rebuttal and observe that __SCoreBO outperforms the other methods on 2 out of 3 tasks__, placing second on the third.
>
> __3)__
> $p(\theta|D)$ is the posterior over the HPs given the data. We can only sample from it through MCMC since it does not have a closed-form expression.
>
>
> __4)__
> $p(f |\theta, D)$ is the distribution over functions that the GP with HPs \theta induces. Step 6 draws a sample function $f_i$ from $p(f |\theta, D)$, then maximizes it to obtain a __simulated optimal location__ and value $(x^*, f^*)$. To obtain $f_i \sim  p(f |\theta, D)$, we use the decoupled sampling approach of Wilson et. al. (2020). When that sample function $f_i$ has been obtained, we use gradient-based optimization to find $({x_i}^*, {f_i}^*)$. The explanation in line 173 will be expanded in the CR.
>
>
> __5)__
> Throughout the paper, the observations are denoted by D = $(x_i, y_{x_i})_{i=1}^N$. The random variable $y_x$ denotes a not-yet-observed random variable (modeled by the GP) at the input location $x$.
> SAL and SCoreBO __do not need to query the black-box function to evaluate the acquisition function__ (that would indeed invalidate all that Bayesian optimization stands for). As mentioned previously, $p(y_x|D)$ is the distribution of the function values at $x$, so (9) evaluates the disagreement between the distributions of the output at $x$.
>
> __6)__
> As $y_x$ is the random variable representing the output at $x$, $p(y_x|D)$ is the predictive posterior distribution of the GP at the input location $x$.
>
> __7)__
> The random variable $f$ is noiseless, whereas $y_x$ is noisy. However, more importantly $p(f|D, \theta)$ is the GP-induced distribution over _functions_, whereas $p(y_x|D, \theta)$ is the predictive distribution of the noisy output _at one specific location_ $x$.
>
> __8)__
> $(x^*, f^*)$ are random variables denoting the optimal value and its location, using the same notation as Hvarfner et. al. (2022). These are samples as described in (4.).
>
> __9)__
> The general trend in high-dimensional tasks is that the model is tailored to take into account the high-dimensional setting, such as in the authors’ AddGPs and SAASBO experiments, but also in methods such as REMBO, ALEBO and BaXuS. In this context, SCoreBO experiments span a comprehensive range of dimensionalities, including 8D, 11D, 25D and 180D.
>
> __10)__
> While the extension of B-QBC into SAL, which is the essence of Eq. (7), in itself is a significant contribution, it is only a building block in the overall body of work. The paper further __presents the first strategy for joint optimization and active learning of HPs__. AL and BO have historically been considered as two different domains, and there has historically been little effort in integrating elements of active learning into BO. SCoreBO proposes a novel method to achieve this by incorporating Entropy Search concepts into the SAL objective. The estimation schemes conditioned on the HPs and the optimizers should also be considered a technical contribution.
>
> The need for the aforementioned joint objective is made clear from the results in Fig. 7 and Fig. 8, where active dimensions of high-dimensional problems must be learned for the optimization to be effective. In Fig. 1 of the rebuttal PDF, we show the HP convergence of SCoreBO and JES for 25D Ackley with 4 active dimensions. Clearly, JES fails to identify the active dimensions, whereas SCoreBO rapidly succeeds. The SCoreBO strategy is fundamentally different from that of JES because it __trades HP learning and optimization in a principled manner__. JES, however, only does the latter.
>
> On the lack of theoretical guarantees, the authors would like to point out that __the JES papers do not provide any convergence guarantees__ (Hvarfner et. al., Tu et. al., 2022), unfortunately. Furthermore, the MES proof (Wang and Jegelka, 2017) is highly contested (Takeno et. al., 2022).
>
>
> ____
>
> Lastly, We thank the reviewer for the suggestions which will be added to the CR. The Wasserstein distance does have a closed form for Gaussians, which we will add as well as suggested.
>
> We hope that the clarifications regarding the contributions and the requested supplementary experiments have enhanced the reviewer's perception of our work. If that is the case, we would appreciate if the reviewer increases their rating. If any ambiguities persist, we are happy to clarify further.

---

> > ### Comment · Reviewer_BCnP · 2023-08-19
> > **Thanks for the clarfications**
> >
> > I would like to express my sincere appreciation to the authors for taking their time to address my previous concerns and clarifying my misunderstandings regarding this paper. I have several followup questions.
> >
> > 1) Could the authors provide clarifications on the highest dimension of the experiments discussed in the paper?
> >
> > 2) With regards to the JES paper authored by Ben Tu et al., I understand that the provided proofs are not intended to establish a definitive convergence rate. But it is good to know some theoretical properties of JES, so is for this paper.
> >
> > 3) Could the authors give intuitive reasons why the proposed method works, and performs better than the other methods?

---

> > > ### Author Response · Authors · 2023-08-20
> > > **Further response**
> > >
> > > Thanks to the reviewer for their continued engagement. We hope to clarify the unique strengths of our proposed method and clarify the additional questions below:
> > >
> > >
> > > ---
> > > > __1.__ Could the authors provide clarifications on the highest dimension of the experiments discussed in the paper?
> > > ---
> > > The highest dimensional benchmark in the experiments is the Lasso-DNA real-world task with 180 variables. The problem constitutes finding the hyperparameters of a weighted Lasso model for a microbiology classification task (Mills 2020). This benchmark has been considered in several papers for high-dimensional hyperparameter optimization (Šehić et al., 2022, Papenmeier et., al. 2022, Ziomek and Bou-Ammar, 2023). As traditional BO methods are inefficient at such high dimensions, we adopt the Sparse Axis-Aligned Subspace hyperparameter prior (SAAS, Eriksson and Jankowiak, 2021) which encourages the model to disregard inactive dimensions. This approach was used for Lasso-DNA in Papenmeier et. al. (2022).
> > > SCoreBO is able to infer the effective subspace under the SAASBO prior within tens of iterations by actively learning the model while optimizing, while conventional acquisition functions fail to infer the effective subspace at all. As such, SCoreBO achieves a 25% performance increase over JES and EI, relative to the initial design configurations in Fig. 7.
> > >
> > >
> > >
> > > ---
> > > > __2.__ With regards to the JES paper authored by Ben Tu et al., I understand that the provided proofs are not intended to establish a definitive convergence rate. But it is good to know some theoretical properties of JES, so is for this paper.
> > > ---
> > > We agree with the reviewer that a deeper theoretical contribution in this area would be very valuable, as the broader set of BO acquisition functions with fully Bayesian treatment of the hyperparameters lack such insights.
> > >
> > > SAL was developed with BQBC in mind, but generalizes BQBC to allow a broader class of disagreement metrics than just the difference in mean. As such, SAL encompasses a broad class of possible AL methods. A theoretical connection with existing methods came through the discussion with reviewer LR3N: SAL with KL divergence used as a distance measure is equivalent to the popular BALD active learning method. We theoretically prove this connection in our discussions with reviewer LR3N.

---

> > > > ### Author Response · Authors · 2023-08-20
> > > > **Further response, part 2**
> > > >
> > > > ---
> > > > > __3.__ Could the authors give intuitive reasons why the proposed method works, and performs better than the other methods?
> > > > ---
> > > > The efficiency of surrogate model-assisted optimization is often highly dependent on that the model yields accurate predictions, which in turn relies on well-specified hyperparameters for the task at hand. The hyperparameters are estimated from the observed data, but __no previous BO method actively selects data to quickly adapt the model to the problem__. Instead, conventional BO methods (EI, UCB, JES) must hope that the selected points happen to be sufficient to learn a good model.
> > > >
> > > > This is where SCoreBO differs from previous approaches, and the reason why SCoreBO outperforms previous methods in the experiments. By selecting points where the model disagrees the most, not only do we quickly learn the hyperparameters, but we __focus on learning the hyperparameters that most strongly impact the model prediction__. By combining this with optimizing the objective, SCoreBO presents a reliable acquisition function that learns the model early and, subsequently, __uses this more accurate model__ to efficiently find good solutions.
> > > >
> > > >
> > > > In the high-dimensional case of Ackley (rebuttal PDF Fig. 1, response to LR3N), SCoreBO identifies the most important dimensions as fast as the pure active learning method BALD, which effectively turns the 25D optimization problem into a 4D one using only tens of iterations. Conventional BO acquisition functions fail on this high-dimensional task, as they do not quantify utility based on any notion of uncertainty in the surrogate. From the hyperparameters in the rebuttal PDF in Fig. 1, the reviewer can see that JES & MES seemingly randomly suggest dimensions as being important. The result is persistent for Hartmann and Lasso-DNA, which shows the potential impact of a joint objective. A counterexample is for the Rosenbrock (4D) synthetic function (Appendix C.3), where SCoreBO performs worse relative to other acquisition functions because the model is already accurate enough. On Rosenbrock, the hyperparameter values increase over time instead of converge, which suggests that the latent function is not part of the model class. Thus, the self-correction effort of SCoreBO is not helpful towards efficient optimization in that case.
> > > >
> > > > SCoreBO is __the first acquisition function to have a joint optimization/active learning objective__, which ultimately makes it very distinct to JES, or any current BO acquisition function. We believe it has an important role to fill in practical applications with complex models, which we have hopefully exemplified both with the SAASBO and HEBO experiments, as well as a theoretical concept to inspire new methods focusing on better learning the surrogate model.

---

> > > > > ### Comment · Reviewer_BCnP · 2023-08-22
> > > > > **Thanks for the clarification**
> > > > >
> > > > > I appreciate the authors' clarifications, and vote for acceptance.

---

### Official Review · Reviewer_LR3N · 2023-07-01

**Soundness:** 2 fair
**Presentation:** 3 good
**Contribution:** 3 good
**Rating:** 6
**Confidence:** 4

**Summary:**

The paper introduces a technique called statistical distance-based active learning (SAL) for the purpose of learning the hyperparameters of a Gaussian process. Additionally, SAL is integrated with information-theoretic Bayesian optimization (BO) in a framework referred to as self-correcting BO (SCoreBO). This integrated approach enables simultaneous learning of GP hyperparameters and optimization of the black box function. The experimental results demonstrate the enhanced performance of SCoreBO compared to traditional benchmarks, including unconventional BO tasks.

**Strengths:**

The paper introduces new active learning algorithms that leverage the Query-by-Committee strategy and utilize two statistical distances. It also addresses the practical challenge of simultaneously learning hyperparameters while optimizing a black-box function, which holds significant importance in real-world applications.

**Weaknesses:**

The main weakness of the paper is inadequate literature review, resulting in major concerns on the novelty of the proposed problem and solution and the experimental results.

1. Although BALD is widely recognized as a highly effective active learning strategy (e.g., employed in PES, MES, JES), it is surprising that BALD is not considered as a baseline in the active learning experiments. Conversely, the other baselines employed in the active learning experiments are relatively weak, such as utilizing only the posterior variance (BALM), the posterior mean (BQBC), or simply adding BALM and BQBC.

2. Equation (7) (the proposed SAL) is essentially the Jensen-Shannon divergence, representing the mutual information between $\theta$ and $y_x$—in other words, BALD when $d$ corresponds to the KL divergence. Surprisingly, this crucial aspect is not discussed in the paper, raising the question of why the preference is given to the Hellinger distance or Wasserstein distance instead of the KL divergence.

3. The paper introduces the problem of optimizing a black box function with unknown Gaussian Process (GP) hyperparameters, asserting that this aspect has received limited attention, without providing references to existing Bayesian Optimization (BO) works that address unknown GP hyperparameters. However, it is worth noting that in the early work PES, there exists a dedicated section that tackles the issue of unknown GP hyperparameters by learning the posterior distribution of the hyperparameters and averaging PES over samples of GP hyperparameters. This crucial detail is missing from the paper, despite frequent mentions of PES throughout. Furthermore, this implies that other information-theoretic acquisition functions such as MES and JES could also be extended to handle unknown GP hyperparameters using a similar approach. Consequently, the novelty in the problem appears to be overstated and a significant concern arises regarding the absence of several essential baselines in the experiments, including PES, MES, and JES averaged over samples of GP hyperparameters.

4. The motivation behind the proposed SCoreBO in Equation (8) needs further enhancement. Given that the goal of Bayesian Optimization (BO) is to optimize a black-box function, it seems appropriate to exclude $\theta$ from the second argument in $d$ in Equation (8). This is because it is redundant to learn the hyperparameters of the Gaussian Process (GP) if they do not impact the difference between $f^*$ or $x^*$ under different GP hyperparameter samples. Essentially, the acquisition function should be a JES, with the GP hyperparameters marginalized out. To substantiate this approach, it would be valuable to present compelling experimental results comparing it with such a JES that marginalizes out the GP hyperparameters.

**Questions:**

Please clarify the above weaknesses.

========================

After reading author's response

The authors have made substantial efforts in providing supplementary experimental outcomes and elaborating the reasoning behind the integration of hyperparameters into the acquisition function.

I believe that these experiments and explanations have the potential to provide stronger motivation for the paper and make the proposed approach's performance more convincing. As a result, I improve my rating accordingly.

**Limitations:**

The authors have adequately addressed the limitations.

---

> ### Author Rebuttal · Authors · 2023-08-08
>
> We thank the reviewer for their through review. There are, however, crucial misunderstandings regarding our work: __all baselines use fully Bayesian hyperparameter treatment__ and the active learning presented in this paper __goes beyond conventional hyperparameter marginalization__. The feedback related to these issues is addressed in __1)__.
>
> We will further clarify the interplay between fully Bayesian hyperparameter treatment and active learning and address the reviewer's remaining concerns. We hope that this gives a fresh perspective on the novelty of the work as perceived by the reviewer.
>
> ---
> > __1)__ The paper __(a)__ lacks _“several essential baselines in the experiments, including PES, MES, and JES averaged over samples of GP hyperparameters”_ as well as __(b)__ lacks references to the _“dedicated section [in PES] that tackles the issue of unknown GP hyperparameters by learning the posterior distribution of the hyperparameters and averaging PES over samples of GP hyperparameters.”_
> ---
>
> __a) All baselines marginalize over the hyperparameters__ in the original manuscript as per Line 224-225:
>
> #### "ScoreBO _and all baselines_ use fully Bayesian hyperparameter treatment."
>
> We will ensure that this is clearer in the CR.
>
> __b)__ The aforementioned Section 2.3 - _Hyperparameter Learning_ in PES simply describes a fully Bayesian treatment of the hyperparameters, i.e. an extension of the work by Osborne et. al. (2010) and Snoek et. al. (2012). These seminal works are referenced on line 27.
>
> However, the aforementioned __fully Bayesian hyperparameter treatment must be distinguished from actively learning the hyperparameters__, i.e. selecting queries to find __more accurate hyperparameters__. Fully Bayesian hyperparameter treatment is a __prerequisite__ for (Bayesian) active learning to take place. Moreover, the active learning component differentiates SCoreBO from PES/JES/MES — SCoreBO learns __more accurate__ hyperparameters (in the spirit of BALD) while optimizing the objective, whereas ES methods - _even their fully Bayesian variants_ - only do the latter.
>
>
> —
> > __2)__: __(a)__ _“[I]t is redundant to learn the hyperparameters of the Gaussian Process (GP) if they do not impact the difference between or under different GP hyperparameter samples”_ and __(b)__ _“Given that the goal of Bayesian Optimization (BO) is to optimize a black-box function, it seems appropriate to exclude theta from the second argument in d in Equation (8)”_. Lastly, __(c)__ _“it would be valuable to present compelling experimental results comparing it with such a JES that marginalizes out the GP hyperparameters.”_
>
> __a.__ Having an __accurate__ sense of the HPs is often crucial for the predictive performance of the GP model and, hence, for the efficiency of the BO algorithm.  To exemplify this, the reviewer is referred to Fig. 1 in the rebuttal PDF, where the impact of the active learning on hyperparameter convergence is visualized for the 25D Ackley task. __Fully Bayesian MES and JES never find the active dimensions__ 1-4, whereas __SCoreBO rapidly finds them__ and outclasses MES/JES/EI as a result (Fig. 7 in main paper). SCoreBO’s joint objective of active hyperparameter learning and optimization makes it __generate more accurate hyperparameters__ by reducing hyperparameter- _and_ optima-induced disagreement. Thus, it quickly obtains hyperparameters which suggests the right dimensions 1-4 as active.
>
> This should clarify that active learning is _not_ redundant in a BO context, as the investment in active learning can produce substantially more accurate hyperparameters, which in turn can yield improved optimization efficiency.
>
> __b.__ The $\theta$ parameter in the second argument of Eq. (8) __enables the active learning component of the joint BO/AL objective - the paper’s main contribution__. For this reason, it is paramount to keep $\theta$ as described. Without $\theta$, SCoreBO would be similar in spirit to fully Bayesian JES.
>
> __c.__ As established previously, the JES in the experiments employs hyperparameter marginalization.
>
>
> ---
> > __3)__ BALD is not considered as a baseline in the active learning experiments.
> ---
> We initially decided to exclude BALD for its lackluster performance in Riis et. al. (2022). However, we agree with the reviewer and added it in the rebuttal PDF in Fig. 2. It performs very well, only marginally worse than SAL-HR on average. We thank the reviewer for pointing this out.
>
> ---
> > __4)__: SAL is essentially the Jensen-Shannon divergence (...)
> ---
> Equation (7) proposes a general instance where the specific distance metric can be instantiated. We have added the Jensen-Shannon (JS) divergence to both the AL and BO experiments in Fig. 1 and Fig. 2 of the rebuttal PDF. It performs very well on some tasks, but does not generally achieve the desired consistency.
>
> The KL/JS divergence is indeed related, but not equivalent, to BALD. Both SAL-KL and SAL-JS minimize the _relative entropy_ between distributions. BALD, which computes the _differential entropy_ minimization, _would_ be equivalent only if the reference distribution was uniform. There is an analogous discussion in Henning and Schuler (2012, p.13).
>
> ____
> We hope that we have clarified the role of the joint optimization and HP learning objective, and the distinct differences between this and previous work. With these misunderstandings out of the way, we would appreciate if the reviewer reconsidered their score. Moreover, we would be happy to address additional questions.
>
> #### References
> 1. Christoffer Riis, Francisco Antunes, Frederik Boe Hüttel, Carlos Lima Azevedo, Francisco Câmara Pereira. Bayesian Active Learning with Fully Bayesian Gaussian Processes. NeurIPS, 2022.
> 2. Phillip Henning and Christian Schuler. Entropy Search for Information-Efficient Global Optimization. JMLR, 2012.

---

> > ### Comment · Reviewer_LR3N · 2023-08-15
> >
> > Thanks to the authors for clarifications and additional experimental results. However, my concerns remain:
> >
> > 1. I believe it is important for the paper to discuss existing BO works that handle the unknown hyperparameters issue such as PES. Considering that the issue of unknown hyperparameter is not new, the novelty of the problem seems to be overstated in the paper.
> >
> > 2. Section 2.3 in PES is not exactly about a fully Bayesian treatment of the hyperparameters. It proposes to average the PES acquisition function over $M$ samples of the hyperparameters (see Equation 10). This is different from a fully Bayesian hyperparameter treatment which marginalizes $p(y|x) \approx 1/M \sum_{i=1}^M p(y|x,\phi_i)$. While this may not be a "correct" way, the paper has shown superior empirical performance. Hence, I do not understand why the experiments did not incorporate this PES baseline.
> >
> > 3. I am still confused why JS divergence is not equivalent to BALD. Let say the posterior distribution of the hyperparameters is approximated with $M$ samples of the hyperparameters, then BALD, i.e., the mutual information between $y$ and the hyperparameters $\phi$ is $I(y;\phi|x) \approx JSD(p(y|x,\phi_1), p(y|x,\phi_2), \dots, p(y|x,\phi_M)) = 1/M \sum_{i=1}^M D(p(y|x, \phi_i) ||  p(y|x))$ where $D$ is the KL divergence. Equation (7) in the paper is this formulation but replacing the KL divergence with other divergences. As the authors demonstrate in their response, the competitive performance of BALD reduces the need of shifting from using the KL divergence to the suggested divergences.
> >
> > 4. I have a different perspective regarding the authors' assertion that: having an accurate hyperparameters is crucial for the efficiency of the BO algorithms. This is due to the fact that obtaining precise hyperparameters requires utilizing samples. If the core objective of BO is to locate maximizers (not to learn the hyperparameters), then what justifies the explicit incorporation of hyperparameters into the acquisition function? I believe we should only reduce the uncertainty in hyperparameters that specifically contributes to uncertainty in the maximizer (which is achieved by a fully Bayesian treatment of PES, for example). Therefore, incorporating the hyperparameters directly into the acquisition function doesn't appear to be particularly persuasive. I understand that there may be an empirical performance gain as the fully Bayesian treatment of the hyperparamters is not exact (approximated with MCMC). As a result, from my personal view, the inclusion of hyperparameters in the acquisition function doesn't seem to be a substantial contribution.

---

> > > ### Author Response · Authors · 2023-08-17
> > > **Further comments on marginalization**
> > >
> > > We'd like to express our sincere appreciation to the reviewer for their consistent engagement throughout the review process and for taking the time to participate in this rebuttal phase. The authors are committed to addressing the concerns raised in the ensuing discussion and especially to clarify the misunderstanding about the marginalization of the hyperparameters.
> > >
> > > ____
> > > __Point 2:__ We now appreciate the misunderstanding of the terminology, which is one that is deeply seeded in the BO community. The reviewer makes  a distinction between
> > > 1. Marginalizing the __acquisition function__ over the hyperparameter uncertainty, $E_\theta[(\alpha(y)|x,\theta)]$, and
> > > 2. Marginalizing the __model__ over the hyperparameter uncertainty, $\alpha(E_\theta[p(y|x,\theta)])$.
> > >
> > > In the ML literature, it'd be natural to call 2) fully Bayesian treatment, which is what the reviewer does. However, the BO community has been referring to a fully Bayesian treatment as directly marginalizing the acquisition function over the hyperparameters, which is 1).
> > > This definition of fully Bayesian treatment in BO was given by Snoek et al. (2012) and used even earlier by Osborne et al. (2010). It has subsequently been used by PES and a number of other BO methods, including MES and JES. From a general ML perspective this may be confusing because the PES authors refer to 1) as a fully Bayesian treatment (see Section 2.3. of their paper). SCoreBO uses 1) as well.
> > >
> > > To summarize, both PES and Snoek refer to the same definition 1) of fully Bayesian treatment, which marginalizes the acquisition function over the hyperparameters. That is the same definition that we use in all the baselines of the paper. This could be misleading and we will make this difference explicit in the CR.
> > >
> > > As a result, since PES performs the same operations to the hyperparameters as other baselines, one would expect that PES should not perform better than the other methods. We have run this version of PES, and added a table of the hyperparameter values for all methods (and reference values) for PES, JES, and SCoreBO on the 25D Ackley task. It is attached as the last comment in this post. We will add PES on all tasks for the CR; it has been run and performs marginally worse than MES on average. Its final performance is added for all tasks as a separate post.
> > >
> > >
> > > ____
> > > __Point 1:__
> > > We agree with the reviewer that the topic of handling unknown hyperparameters for BO is not new — All optimizers must handle unknown hyperparameters in some way, whether through MLE, MAP, fully Bayesian treatment or adapting the acquisition function to actively manage uncertainty across hyperparameters.
> > >
> > > Since the reviewer is familiar with BALD and PES, we would like to highlight the differences between them to align with the reviewer:
> > > - PES reduces the uncertainty over the optimum,
> > > - BALD reduces the uncertainty over the hyperparameters, and
> > > - SCoreBO reduces uncertainty over optimum **and** hyperparameters.
> > >
> > > The authors would like to emphasize that a Bayesian treatment of the hyperparameters does not entail reducing hyperparameter uncertainty, i.e., a Bayesian treatment “acknowledges” parameters uncertainty, but doesn’t reduce it. It can still suffer from having large variance in the parameter distribution — That is a crucial difference that makes SCoreBO outperform other acquisition functions on the SAASBO experiments.
> > >
> > > Furthermore, the Bayesian treatment of the hyperparameters in PES is based on Snoek et. al. (2012) and it is not a contribution of PES. PES is one of the many approaches that use marginalization of the hyperparameters in this way (MES and JES also did that, following Snoek et al and PES). We use the same marginalization over the hyperparameters used in PES, MES and JES. Additionally, we cite and compare against all these baselines. We thus disagree with the statement that we do not discuss existing BO works that handle unknown hyperparameters.
> > >
> > > The main contribution of SCoreBO is to combine the reduction over the uncertainty of the location of the optimum, such as in PES, and the reduction of the uncertainty over the hyperparameters. This __joint__ uncertainty reduction is novel.
> > >
> > > We are committed to improving the introduction of the paper in the camera ready, to make sure that our contribution is not confused with general handling of unknown hyperparameters as the reviewer suggests.

---

> > > > ### Author Response · Authors · 2023-08-17
> > > > **Further comments on BALD equivalence**
> > > >
> > > > __Point 3:__ The reviewer is correct in their assessment, and we are __greatly thankful__ for their persistence!
> > > > We note that the equality does not only hold in the finite-sample MC approximation that the reviewer put forward, but also in the general case, i.e. the JSD for an infinite number of distributions. This was an oversight on our end, and we believe it provides another nice link to existing methods, such as  BQBC. We will happily include this relation, and the short proof, in the CR. With this help by the reviewer, we believe that SAL becomes a convenient umbrella term for multiple existing methods. Naturally, this means that there is a corresponding SCoreBO variant using KL-divergence/BALD, which we are happy to include as well. We will acknowledge the reviewer's contribution in formalizing the explanation of the methods in the acknowledgement section of the CR.
> > > >
> > > > We will also run the reverse SAL-KL (since BALD is already included) and provide the results. The proof for the aforementioned equivalence is provided in the end of our post.
> > > >
> > > > ____
> > > >
> > > > __Point 4:__ The reviewer raises a valid point regarding the motivation behind the formulation of SCoreBO in Equation (8). While the aim of Bayesian Optimization (BO) is indeed to efficiently optimize a black-box function, we would like to emphasize that accurate hyperparameter estimation can significantly impact statistical efficiency.
> > > > The experiments reveal that in scenarios with inaccurate hyperparameter distribution, optimizing the objective directly can be inefficient (please refer to the fully Bayesian treatment & BoTorch prior in Fig. 1 of the paper). In such cases, the lack of a precise model of the response surface hinders effective BO. By dedicating iterations to better understand the problem, even though it initially incurs a cost, we observe improved optimization performance later on.
> > > > It's important to note that we do not claim that this trade-off always guarantees to be beneficial. Our intention is to demonstrate instances where it proves advantageous and a few instances where it doesn't. For instance, in Fig. 1 of the rebuttal PDF, we highlight a case where the posterior distribution is notably imprecise, rendering the sampled points non-representative of the underlying function, as the posterior distribution at large suggests that all dimensions are unimportant.
> > > >
> > > > Prior to the introduction of SCoreBO, the ability to deliberately allocate attention to active hyperparameter learning for enhanced optimization performance was lacking. This underscores a significant contribution of our paper.
> > > >
> > > > ____
> > > >
> > > > We remain open to any additional questions, suggestions, or concerns that may arise as we proceed in this interactive dialogue. Your comments have been immensely helpful in guiding our responses and enhancing the clarity of our contributions. We look forward to further productive exchanges.

---

> > > > > ### Author Response · Authors · 2023-08-17
> > > > > **BALD equivalence proof**
> > > > >
> > > > >  $$
> > > > > BALD = I(y,\theta) = H(p(y)) - E_{\theta}[H(p(y,|\theta))]=
> > > > > $$
> > > > > $$
> > > > > -\int_{-\infty}^{\infty}\int_{\theta} p(\theta)p(y|\theta) \log[p(y)] d\theta dy + \int_{\theta} p(\theta)\int_{-\infty}^{\infty} p(y|\theta)\log [p(y|\theta)]dy d\theta =
> > > > > $$
> > > > > $$
> > > > > \int_{\theta} p(\theta)\int_{-\infty}^{\infty} p(y|\theta) \log\left[\frac{p(y|\theta)}{p(y)}\right]dy d\theta = \int_{\theta} p(\theta) KL(p(y|\theta) || p(y)) d\theta
> > > > > $$
> > > > > $$
> > > > > = E_{\theta}[KL(p(y|\theta)||p(y))] = SAL.KL
> > > > > $$

---

> > > > > > ### Author Response · Authors · 2023-08-17
> > > > > > **FB-PES & BALD Hyperparameter convergence**
> > > > > >
> > > > > > The below table shows the (log) hyperparameter values (mean and standard deviation across the samples in the FB treatment) for all methods after 100 iterations of 25-dimensional Ackley with 4 active dimensions, across 10 repetitions.
> > > > > >
> > > > > > $\ell_i$ close to -1 suggests that the dimension $i$ is active, and very important to optimization performance. This is true for $\ell_1, \ldots, \ell_4$. The inactive dimensions $\ell_5, \ell_6$ should have a confident and large value. The true noise $\sigma_\epsilon^2$ is $-1$.
> > > > > >
> > > > > > Fully Bayesian PES is and displays inaccurate and inconfident values for all parameters, as seen previously with MES, JES and EI. BALD (previously not in plot) produces accurate values since it actively learns the hyperparameters.  However, __SCoreBO is even more accurate, and confident, on all parameters__ despite doing optimization at the same time. PES subsequently produces similar to JES and MES on the task, since it does not know which dimensions to explore.
> > > > > >
> > > > > > |                     | JES   | MES     | ScoreBO        | PES        | BALD       |
> > > > > > |:--------------------|:------------|:-----------|:---------------|:-----------|:-----------|
> > > > > > | $\sigma_\epsilon^2$ | -1.73±0.73  | -2.37±0.94 | -0.99±0.28     | -1.97±0.72 | -1.32±0.42 |
> > > > > > | $\ell_1$            | 0.11±0.92   | 0.7±0.84   | -0.89±0.1      | 0.46±1.42  | -0.68±0.35 |
> > > > > > | $\ell_2$            | 0.24±1.02   | 0.46±1.04  | -0.89±0.1      | 0.86±1.15  | -0.75±0.49 |
> > > > > > | $\ell_3$            | -0.02±1.29  | 0.45±1.15  | -0.89±0.1      | 0.45±1.32  | -0.71±0.29 |
> > > > > > | $\ell_4$            | 0.66±0.78   | 0.68±0.9   | -0.88±0.12     | 1.05±0.93  | -0.68±0.32 |
> > > > > > | $\ell_5$            | 0.8±0.64    | 1.0±0.71   | 1.01±0.3       | 1.28±0.62  | 0.74±0.69  |
> > > > > > | $\ell_6$            | 0.7±0.82    | 0.5±1.06   | 1.01±0.31      | 0.97±1.15  | 0.69±0.69  |

---

> > > > > > > ### Author Response · Authors · 2023-08-17
> > > > > > > **Synthetic BO & DL performance**
> > > > > > >
> > > > > > > ### Novel results for fully Bayesian PES only
> > > > > > >
> > > > > > > Log regret on Synthetic BO tasks (mean and standard error) including PES and previous methods. PES performs worst on three tasks and best on one. SCoreBO wins on four tasks. All methods employ fully Bayesian acquiisition function treatment (i.e. Section 2.3 in PES)
> > > > > > > |              | Branin     | Hartmann (3D)   | Hartmann (4D)   | Hartmann (6D)   | Rosenbrock (2D)   | Rosenbrock (4D)   |
> > > > > > > |:-------------|:-----------|:------------|:------------|:------------|:--------------|:--------------|
> > > > > > > | NEI          | -1.64±0.06 | -1.4±0.05   | -1.07±0.06  | -0.6±0.06   | -0.92±0.07    | -0.22±0.05    |
> > > > > > > | JES    | -1.84±0.08 | __-1.62±0.06__  | -1.26±0.07  | -0.88±0.05  | -1.07±0.08    | -0.29±0.05    |
> > > > > > > | MES       | -1.89±0.09 | -1.58±0.06  | -1.03±0.07  | -0.78±0.08  | -1.0±0.07     | -0.34±0.06    |
> > > > > > > | ScoreBO | __-2.11±0.08__ | -1.56±0.06  | __-1.34±0.07__  | __-1.06±0.05__  | __-1.23±0.08__    | -0.16±0.06    |
> > > > > > > | PES          | -1.9±0.09  | -1.24±0.07  | -0.94±0.06  | -0.47±0.08  | -1.0±0.09     | __-0.44±0.06__    |
> > > > > > >
> > > > > > > Performance on deep learning HPO tasks (mean and standard error) including PES and previous methods. fully Bayesian PES (i.e. Section 2.3 execution) performs approximately on par with fully Bayesian MES as the second-worst or worst method.
> > > > > > >
> > > > > > > |              | WMT    | CIFAR   | LM1B   |
> > > > > > > |:-------------|:-----------|:------------|:-----------|
> > > > > > > | NEI          | 29.1±0.16  | 18.45±0.12  | __60.32±0.14__ |
> > > > > > > | JES    | 29.04±0.14 | 17.82±0.17  | 60.53±0.13 |
> > > > > > > | MES       | 29.63±0.16 | 18.57±0.11  | 60.57±0.13 |
> > > > > > > | ScoreBO |  __28.82±0.12__ | __17.77±0.14__  | 60.42±0.08 |
> > > > > > > | PES          | 29.21±0.16 | 18.01±0.1   | 60.57±0.09 |

---

> > > > > > > > ### Comment · Reviewer_LR3N · 2023-08-20
> > > > > > > >
> > > > > > > > While my concern regarding the novelty has not been entirely addressed, I deeply value the authors' efforts to conduct additional experiments and address my inquiries. I hope that the authors will share the code to aid other researchers working on similar subject. I have improved my score accordingly.

---

> > > > > > > > > ### Author Response · Authors · 2023-08-20
> > > > > > > > > **Thanks**
> > > > > > > > >
> > > > > > > > > We greatly appreciate the reviewer's efforts, as they have undeniably improved the quality of our paper through the SAL-KL discussion and beyond.
> > > > > > > > >
> > > > > > > > > Thank you for your remark about sharing code to aid other researchers working on similar subjects. We fully agree that this is very important to the community. While the program chairs have instructed authors to not post URLs during the rebuttal period, we invite the reviewer to check the URL for our public anonymous repository linked in our original paper on page 7, Lines 226-227, which includes complete code for our method and all baselines in BoTorch (recently updated to include all the experiments and methods added during the rebuttal), as well as thorough instructions and scripts for all experiments in the paper, which produce all the required metrics (MLL, observed values, inference values, hyperparameter sets) to reproduce our work.
> > > > > > > > >
> > > > > > > > > If the reviewer has any additional comments or questions following our most recent set of responses, we would be keen to address them with the remaining time until the end of the rebuttal phase.

---

### Official Review · Reviewer_fp2C · 2023-07-06

**Soundness:** 3 good
**Presentation:** 3 good
**Contribution:** 3 good
**Rating:** 6
**Confidence:** 3

**Summary:**

The paper presents two algorithms/methods, SAL and SCoreBO, for active learning and Bayesian optimization using GPs. Both build on the premise that hyperparameters are critical to effective AL or BO; hence it is important to learn both the function and the hyperparameters through an acquisition function that accounts for both needs.

Contributions:
- The SAL acquisitions function for AL considers the statistical distance between a condition (on the hyperparameters) and the marginal posterior.
- The SCoreBO algorithm for BO, which uses SAL in combination with Thompson sampling/posterior sampling
- Empirical evaluation on several relevant benchmark problems
- Comparison with a set of alternative methods



**Strengths:**

- Well-written and clear (with only a few unclear aspects)
- Relatively simple but effective approach  (based on various other works in fully Bayesian BO including [42])
- Thorough evaluation of a suitable set of both AL and BO benchmarks, including non-standard problems
- Comparison with what appears to be a sensible set of baselines, showing favorable performance on the BO tasks

**Weaknesses:**


The following are mostly just questions and comments, not all weaknesses per se:

- Lack of details regarding a fully Bayesian treatment
  -   The paper mentions that it applies a fully Bayesian treatment, yet I am (as a reader) still a bit unsure about the specific inference being used. I think more details are needed here to make it precise. In particular, discussing $p(\theta | D)$ would be helpful to ensure the paper presents a self-contained view.

- Justification of distance metrics and alternatives
  -  I feel the paper lacks justification for the choice of distance measures. I appreciate a choice needs to be made at some point, but alternative statistical distance measures also have closed-form, e.g., (symmetric) KL, and even approximations for mixtures.
  -  I am slightly concerned about the sensitivity to the distance function for the AL (and BO as shown in the appendix). For the AL task, have the authors performed longer experiments than presented in Figure 6 (i.e. why stop at 100/150/200 iterations)?

- What's the complexity? It would be helpful with an indication/summary of the complexity of SCoreBO relative to a standard fully Bayesian BO approach.

- (The need for) theoretical bounds: What is the prospect of providing theoretical bounds for the specific algorithm?

Minor:
- As a sanity check, have the authors compared SCoreBO with a box standard BO approach without a fully Bayesian treatment of hyperparameters (i.e., ML-II)?
- Figure 1: For clarity, I'd suggest specifying what a "BoTorch prior" is
- Figure 7: There are two dashed lines; perhaps clarify the caption.




**Questions:**

Included in the above.

**Limitations:**

Included in the above.

---

> ### Author Rebuttal · Authors · 2023-08-08
>
> We thank the reviewer for their feedback and are glad to see that the reviewer appreciates both the importance of the problem setting as well as the proposed approach.
>
> ---
> > __1)__ _Lack of details regarding a fully Bayesian treatment_
> ---
> We agree with the reviewer that the paper would benefit from more clearly describing the sampling procedure when marginalizing over the hyperparameters. In the CR, we will extend the description of this in Appendix A1 as well as add a shorter description in the main text.
> Briefly, SCoreBO uses No U-Turn Sampling (NUTS) MCMC sampling (the Pyro implementation) and marginalizes over 16 models.
>
> We will allocate a section of the background to fully Bayesian hyperparameter treatment in the CR.
>
> ---
> > __2)__ _Justification of distance metrics and alternatives_
> ---
>
> The requested Jensen-Shannon (JS, symmetric KL) divergence has been added to the set of AL and BO experiments (Fig. 2, 3 in the attached rebuttal PDF) and __all AL experiments have been doubled in length__. The rank of the different methods is roughly the same as in the original manuscript — We find that __the JS variant is occasionally a powerful alternative but less consistent__ than the Hellinger distance for both AL and BO.
>
> All distance measures have their specific strengths and weaknesses. However, the Hellinger distance has been the most empirically consistent choice overall. This may be due to the intuition below which aligns well with the global optimization goal under limited evaluation budget:
> - The Jensen-Shannon divergence prioritizes __same order-of-magnitude variances__. As such, it is inclined to repetitively query the same location to correctly estimate noise levels.
> - Wasserstein (earth-mover) distance seeks to minimize the difference in __first and second moments__.  In practice, this places a premium on matching large-variance regions, leading to higher global exploration which can be detrimental for global optimization.
> - Hellinger distance seeks to minimize the __ratio between difference in mean and the sum of variances__, which punishes outlier predictive distributions of high confidence. This turns out to be the most practical metric for posterior convergence.
>
> We will clarify these points which characterize the different distance metrics in the CR.
>
>
> ---
> > __3)__ _What’s the complexity? It would be helpful with an indication/summary of the complexity of SCoreBO relative to a standard fully Bayesian BO approach._
> ---
>
> The conditioning on Thompson samples involves a rank-1 update of $\mathcal{O}(n^2)$ of the GP for each Thompson sample draw. As such, the complexity of constructing the acquisition functions is $\mathcal{O}(MNn^2)$ for $M$ models, $N$ optima per model and $n$ data points. The MCMC involved with the fully Bayesian treatment is $\mathcal{O}(|\theta|n^3)$ per sample.
>
> The complexity of the forward pass is the same as (fully Bayesian) JES, namely $\mathcal{O}(MNn^2)$. As such, these methods are roughly identical in terms of runtime. For reference, EI has a forward pass complexity of $\mathcal{O}(Mn^2)$ and no setup. For all acquisition functions, the NUTS sampling accounts for a large majority of the total runtime.
>
> We will extend the complexity section for the CR.
>
> ---
> > __4)__ _(The need for) theoretical bounds: What is the prospect of providing theoretical bounds for the specific algorithm?_
> ---
> Theoretical bounds for SCoreBO are likely tied to bounds for BO algorithms with fully Bayesian treatment, which, to the best of the authors' knowledge, there are none to date. Furthermore, bounds on SCoreBO are likely tied to bounds on to information-theoretic acquisition functions. Among these acquisition functions, PES/JES have no convergence bounds, and the bounds proposed in MES are not believed to be correct (Takeno et. al 2022, App. H). Thus, while theoretical bounds for SCoreBO are an interesting direction for future research, they appear challenging at present.
>
> ---
> > __5)__ _The SCoreBO algorithm for BO, which uses SAL in combination with Thompson sampling/posterior sampling._
> ---
> To ensure clarity: SCoreBO employs JES-like conditioning, which uses Thompson sampling (TS) as an intermediate step, to build the acquisition function. SCoreBO does not, though, use TS itself as the acquisition function.
>
> ____
>
> We hope that the additional results and answers have addressed the reviewer's concerns and improved the reviewer’s perception of our work. We would be happy to address any additional questions that may arise.
>
> #### References
> Shion Takeno, Tomoyuki Tamura, Kazuki Shitara, Masayuki Karasuyama. Sequential and Parallel Constrained Max-value Entropy Search via Information Lower Bound. Proceedings of the 39th International Conference on Machine Learning, PMLR 162:20960-20986, 2022.

---

> > ### Comment · Reviewer_fp2C · 2023-08-18
> >
> > Thanks to the authors for addressing my questions, in particular on the details of the fully Bayesian treatment and choice of divergence. I note these aspects have been discussed further and in-depth with reviewer LR3N, including explicit links to BALD, etc. I welcome the insights gained from the discussion with LR3N (and other comments). While I still feel the paper addresses a relevant and interesting aspect of BO, the discussion reveals that a more complete and coherent narrative is required to present the method/results than the case in the submitted paper. The many required/suggested changes lead to some doubt about what the final paper will look like, and I usually lean towards recommending the paper go through a full review cycle in such cases. I’d encourage the authors to summarize their proposed changes in a single place, before the end of the discussion phase on the 21st.

---

### Official Review · Reviewer_ta8V · 2023-07-21

**Soundness:** 2 fair
**Presentation:** 3 good
**Contribution:** 3 good
**Rating:** 6
**Confidence:** 4

**Summary:**

Standard acquisition functions in Bayesian Optimization (BO) only aim to finding the optimum, but do not directly consider the problem of hyperparameter learning in Gaussian Processes (GP) which has considerable impact on the optimization performance.
This paper introduces a new acquisition function for Active Learning (AL) and Bayesian Optimization (BO), whose goal is to learn both hyperparameters and the location of the optimum. Specifically, the acquisition function for AL is based on extending similar previous proposals of Active Learning via disagreement, by using different statistical distances. This new acquisition function is then adapted for the BO task by conditioning on sampled locations/value of the optimum. The new acquisition functions are shown to work (slightly) better at AL and BO, especially with unusual BO tasks.

**Strengths:**

- **Originality:** While natural, the problem of hyperparameter learning in BO has rarely been addressed, so this paper tackles an open and underestimated problem in the field. The proposed solutions are interesting.
- **Significance:** Given the prominence of BO in machine learning and other fields, this work is potentially very significant.
- **Quality:** The general quality of the work is good, although there are some open questions (see below).
- **Clarity:** The paper is generally clear.

As general comments, Statistical distance-based Active Learning (SAL) is well-motivated and explained, a nice generalization of previous proposals. It is also nice that the paper explores the use of two different distances (Wasserstein and Hellinger), and that it gives approximations on how to compute them. The proposed "motivated heuristic" for extending SAL to BO via conditioning over the location-value of the optimum is interesting.

### Post-rebuttal

Thanks to the authors for addressing the points I raised. I am glad that they managed to run additional experiments with a more realistic benchmark, showcasing the effectiveness of their method. Overall, I am satisfied with the paper and the score of 6 reflects my current evaluation of the paper ("Technically solid, moderate-to-high impact paper, with no major concerns with respect to evaluation, resources, reproducibility, ethical considerations.").

**Weaknesses:**

- The paper showcases many synthetic functions but there are only a couple of real-world applications. Real-world applications are particularly important for this work because they are exactly the situations at risk of breaking active learning (e.g., due to model misspecification). Indeed, the Cosmological Constant task in the paper is the one in which ScoreBO does not show any advantage over vanilla Expected Improvement. For this reason, it would have been nice to show other real examples in which ScoreBO does improve performance.
- As a minor comment, please double-check the paper and bibliography for typos or errors. At a quick glance I spotted a few mistakes (e.g., the authors of reference [43] "Fast information-theoretic Bayesian optimisation" are in a wrong order; occasionally "Bayesian" appears lowercase, etc.).

**Questions:**

- This paper would strongly benefit from showing ScoreBO at work on other real-world tasks, to showcase that indeed this new acquisition function and its focus on active learning does not harm when deployed on real problems (e.g.,  RL, hyperparameter tuning, and other typical applications of BO which are conspicuously missing here).
- Minor: Fix the few typos in the paper and bibliography.

**Limitations:**

The conclusion quickly states a few limitations but it'd be good to have a separate **Limitations** section which very explicitly mentions them.

---

> ### Author Rebuttal · Authors · 2023-08-08
>
> We thank the reviewer for their feedback. We are pleased to see that the reviewer recognizes the significance of the joint BO/hyperparameter learning problem setting and our proposed approach.
>
>
> ---
> > __1.__ _This paper would strongly benefit from showing ScoreBO at work on other real-world tasks, to showcase that indeed this new acquisition function and its focus on active learning does not harm when deployed on real problems (e.g., RL, hyperparameter tuning, and other typical applications of BO which are conspicuously missing here)._
> ---
>
> Three (4D) hyperparameter tuning tasks from the PD1 benchmarking suite have been added in the rebuttal PDF, Fig. 4 — Two involving large language models, and one from computer vision. The surrogate model from HEBO [Cowen-Rivers et. al. 2020], which employs input and output warpings [Snoek et, al 2014], is used to account for the heteroskedasticity prevalent in HPO tasks. SCoreBO outperforms the other methods on 2 out of 3 tasks, placing second on the third.
>
> The three additional real-world benchmarks hopefully depict a clearer picture of the performance of SCoreBO, together with the other two real-world benchmarks included in the original submission, namely Lasso-DNA and Cosmological Constants.
>
>
> ---
> > 2. _The conclusion quickly states a few limitations but it’d be good to have a separate
> Limitations section which very explicitly mentions them._
> ---
>
> The authors would like to thank the reviewer for this suggestion. A dedicated Limitations section will be added in the CR. We intend to discuss the potential pitfalls of utilizing SCoreBO with misspecified models. This is partly addressed in Appendix C in relation to the Rosenbrock functions, where SCoreBO performs worse than on other tasks, relative to other acquisition functions. On Rosenbrock, the hyperparameter values increase over time instead of converge, which suggests that the latent function is not part of the model class. Thus, the self-correction effort of SCoreBO is less rewarding.
>
> The authors believe that these limitations, as well as the broader subject of model misspecification in BO, necessitate further research. Encouragingly, the reviewer's perspective appears to align with this viewpoint.
>
>
> We thank the reviewer for pointing out the typos, which will be addressed in the CR.
>
> _______
>
> Hopefully, the additional real-world applications introduced in this rebuttal showcase the potential and usefulness of SCoreBO. We would be happy to address additional questions that the reviewer might have.

---

> > ### Comment · Reviewer_ta8V · 2023-08-10
> > **Response to the rebuttal**
> >
> > Thanks to the authors for addressing the points I raised. I am glad that they managed to run additional experiments with a more realistic benchmark, showcasing the effectiveness of their method. Overall, I am satisfied with the paper. In the discussion with the other reviewers, I will argue for acceptance.

---

> > > ### Author Response · Authors · 2023-08-10
> > > **Additional reponse to ta8V**
> > >
> > >  We are pleased that the reviewer valued our additional experiments and thankful for their willingness to support our paper in discussions with other reviewers.
> > >
> > > In light of the reviewer’s updated judgement, we would very much appreciate if they also considered updating their score on OpenReview.

---

### Author Rebuttal · Authors · 2023-08-08

The authors would like to thank all reviewers for their effort. As per the reviewers' requests, we have added 4 additional plots to the rebuttal PDF.

__Fig. 1:__ Hyperparameter convergence of fully Bayesian JES, MES and SCoreBO on the SAASBO task, the noisy 25-D Ackley function. We see that the fully Bayesian information-theoretic acquisition functions fail to find any of the active dimensions of the task, while SCoreBO finds all of them with low uncertainty. Thus, SCoreBO successfully optimizes the task, whereas JES and MES do not.

__Fig. 2:__ Prolonged active learning results with double the iteration budget of the initial submission. We include SAL using Jensen-Shannon (JS) divergence as the distance metric, and BALD as an additional baseline. BALD is a highly competitive baseline. SAL-JS performs well on many tasks, but is inconsistent.

__Fig. 3:__ BO synthetic experiments with SCoreBO using Jensen-Shannon divergence as well as non-fully Bayesian EI (MAP). SCoreBO-JS performs well on many tasks, but is inconsistent. EI-MAP performs well on some tasks, but lags behind on some tasks, most prominently on Hartmann (6D).

__Fig. 4:__ Three (4D) hyperparameter tuning tasks from the PD1 benchmarking suite - Two involving large language models, and one from computer vision. SCoreBO outperforms the other methods on 2 out of 3 tasks, placing second on the third.

_________

The authors look forward to a productive discussion with the reviewers!

---

### Author Response · Authors · 2023-08-20
**Summary of changes**

We thank the reviewers for their further engagement. We acknowledge that additional aspects of the SAL/SCoreBO have been brought to light during the rebuttal, and the comments regarding BALD from LR3N have been very helpful to the quality of our work. To summarize, the following is currently being worked into the camera ready version of our paper:

1. __SAL-KL/BALD equivalence:__ An expanded discussion on BALD in the background. The definition of SAL-KL and the proof of its equivalence to BALD, stated in Section 3.1 and proved in the Appendix. (Presented in discussion with __LR3N__, SCoreBO-KL in a subsequent post).
2. __Intuition for distance metrics:__ Clarification of the benefits and drawbacks of various distance metrics, including plots to match the intuition in the Appendix (__fp2C__).
3. __Deep learning benchmarks:__ Inclusion of the tasks, employing the warpings from HEBO, from the rebuttal PDF as an additional Section 4.4. (Rebuttal PDF, Fig 4.)
4. __Details regarding fully Bayesian treatment:__ A paragraph in Section 2 outlining it generally and its use in acquisition functions specifically. (__LR3N__). An extended explanation of the sampling procedure for the fully Bayesian treatment in the Appendix and a shorter description in Section 3.3. (__fp2C__).
5. __Complexity of method:__ A description of the complexity of SCoreBO and fully Bayesian treatment in Section 3.2 as compared to other methods (__fp2C__).
6. __Explicit limitations section:__ A new Section 5.1 outlining the pitfalls of using SCoreBO for misspecified models, and its dependence on fully Bayesian treatment of the hyperparameters (__ta8V__).


Minor (non-structural):
- Fix references and typos (__ta8V__, __BCnP__).
- Include EI-MAP (fp2C) and fully Bayesian PES (__LR3N__).
- Show prolonged AL experiments (__fp2C__).
- Specifying BoTorch prior (currently in Appendix 1, __fp2C__).
- Clarify meaning of dashed lines (__fp2C__).

We note that almost all experiments, figures and commentary _can already be found in this rebuttal_. As such, we hope that the reviewers feel confident that the CR will be representative of the material already presented, and result in an even higher-quality paper. We are currently working on the CR and are happy to provide additional details on any of these bullets if the reviewer desires.

---

> ### Author Response · Authors · 2023-08-20
> **SCoreBO-KL**
>
> To complete the set of experiments, we have now added SCoreBO-KL to the synthetic benchmarks. SCoreBO-KL displays very competitive performance, ranking as the second best method overall (after SCoreBO-Hellinger = SCoreBO-HR), achieving the best regret on one task and the second best on an additional three.  Best __log regret__ on each task is bolded and cursive, second best is bolded only.
>
>
>
> |     		 | Branin     | Hartmann (3D)   | Hartmann (4D)   | Hartmann (6D)   | Rosenbrock (2D)   | Rosenbrock (4D)   |
> |:-------------|:-----------|:------------|:------------|:------------|:--------------|:--------------|
> | NEI 		 | -1.64±0.06 | -1.4±0.05   | -1.07±0.06  | -0.6±0.06   | -0.92±0.07    | -0.22±0.05    |
> | JES    | -1.84±0.08 | __-1.62±0.06__  | __-1.26±0.07__  | -0.88±0.05  | __-1.07±0.08__    | -0.29±0.05    |
> | MES 	 | -1.89±0.09 | -1.58±0.06  | -1.03±0.07  | -0.78±0.08  | -1.0±0.07     | ___-0.34±0.06___    |
> | ScoreBO-KL | __-2.05±0.1__  | ___-1.68±0.06___  | -1.08±0.07  | __-0.92±0.06__  | -0.98±0.08    | __-0.32±0.06__    |
> | ScoreBO-HR | ___-2.11±0.08___ | -1.56±0.06  | ___-1.34±0.07___  | ___-1.06±0.05___  | ___-1.23±0.08___    | -0.16±0.06    |

---

> > ### Author Response · Authors · 2023-08-21
> > **All intended changes**
> >
> > We thank the reviewers for their engagement — This message is a follow-up to the camera ready summary (the message to all reviewers on August 19). The aspects that have been brought to light during the rebuttal are very helpful to the quality of this work and we are grateful to the reviewers. We fully agree with reviewer fp2C that it’s helpful to see the intended revisions in order to make a confident assessment, and while we‘re not allowed to upload the final revised paper we would like to do the next best thing and describe the changes related to the enumerated updates in detail. We also make explicit reference to the line in the current OpenReview PDF where we will add this text.
> >
> >
> > ### 1. SAL-KL/BALD equivalence:
> > __Line 103:__
> > Bayesian Active Learning by Disagreement (BALD) [21] was among the first Bayesian active learning approaches to explicitly focus on learning the model hyperparameters:
> > $$\alpha_{BALD} (x) = I(y_x; \theta) = H(p(y_x)) - E_{\theta} [H(p(y_x | \theta))]$$
> >
> > __Line 147:__ Notably, SAL generalizes both BQBC and BALD, which are exactly recovered by choosing the metric $d$ to the difference in mean or the forward KL divergence, respectively.
> >
> >
> > __Proposition 1:__  $E_\theta [KL(p(y_x|\theta, {D}) || p(y_x|{D}))] = I(y_x; \theta |D).$
> >
> > The proof is shown in Appendix E.
> >
> > __Appendix E - SAL-KL equivalence with BALD:__
> >
> > We show that SAL, using the KL divergence as the metric $d$, is equivalent to BALD.
> >
> > _Proof_:
> > $$
> > BALD = I(y_x,\theta) = H(p(y_x)) - E_{\theta}[H(p(y_x|\theta))]=
> > $$
> > $$
> > -\int_{-\infty}^{\infty}\int_{\theta} p(\theta)p(y_x|\theta)~log[p(y_x)] d\theta dy_x + \int_{\theta} p(\theta)\int_{-\infty}^{\infty} p(y_x|\theta) log [p(y_x|\theta)]dy_x d\theta =
> > $$
> > $$
> > \int_{\theta} p(\theta)\int_{-\infty}^{\infty} p(y_x|\theta) log\left[\frac{p(y_x|\theta)}{p(y_x)}\right]dy_x d\theta = \int_{\theta} p(\theta) KL(p(y_x|\theta) || p(y_x)) d\theta
> > $$
> > $$
> > = E_{\theta}[KL(p(y_x|\theta)||p(y_x))] = SAL.KL
> > $$
> >
> >
> > ### 2. Intuition for distance metrics:
> >
> > __Line 117:__ We focus on three metrics which have closed forms for Gaussian random variables.
> >
> > __Line 119:__
> > The KL divergence is a standard asymmetrical measure for dissimilarity between probability distributions. For two probability distributions $P$ and $Q$, it is given by
> > $$
> > D_{KL} (P || Q)= \int_{X}P(x)log(P(x)/Q(x))dx.
> > $$
> >  The KL divergence mainly prioritizes same order-of-magnitude variances, and will initially query the same location multiple times to assess noise levels. Thereafter, it tends to query in an axis-aligned fashion, close to previous queries, to attain information regarding the length scales, but places a low priority on global exploration.
> >
> > __Line 124:__ The Hellinger distance seeks to minimize the ratio between difference in mean and the sum of variances, which punishes outlier predictive distributions of high confidence. Similar to KL, initial queries have a tendency to be axis-aligned to attain selective length scale information.
> >
> > __Line 127:__ The Wasserstein distance, also commonly referred to as the earth-mover distance, seeks to minimize the difference in the first and second moment. In practice, this places a premium on matching large-variance regions, leading to higher global exploration which can be detrimental for global optimization.
> >
> >
> > ### 3. Deep learning benchmarks:
> > __Page 8:__ We will add a new figure with the new DL results as seen in the rebuttal PDF. We will tentatively call the new figure Figure 10.
> >
> > __Line 285:__ We show the performance of SCoreBO on three (4D) hyperparameter tuning tasks from the PD1 benchmarking suite [Wang et. al., 2023], Fig. 10 — Two involving large language models, and one from computer vision. We adopt the same application setting as in HEBO [Cowen-Rivers et. al. 2020], where input and output warpings [Snoek et, al 2014] are used to account for the heteroskedasticity prevalent in real HPO tasks. We observe that SCoreBO outperforms the other methods on 2 out of 3 tasks, placing second on the third.

---

> > > ### Author Response · Authors · 2023-08-21
> > > **All intended changes, part 2**
> > >
> > > ### 4. Details regarding fully Bayesian treatment
> > >
> > > __Line 80:__
> > > The posterior probability of observing a value $y_x$ for a point $x$ is given as [Lalchand \& Rasmussen, 2020]:
> > >
> > > $$
> > >     p(y_x|D) = \int_{\theta} \int_{f} p(y_x|f, \theta)p(f| \theta, D) p(\theta| {D})df d\theta
> > > $$
> > > where ${D}$ is the observed data. The inner integral is equal to the GP predictive posterior,
> > >
> > > $$
> > >     \int_{f} p(y_x|f, \theta)p(f|\theta, {D})df = p(y_x|{D}, \theta).
> > > $$
> > > However, the outer integral is intractable and is estimated using Markov Chain Monte Carlo (MCMC) methods. The resulting posterior prediction
> > >
> > > $$
> > > p(y_x|{D}) = \int_{\theta}p(y_x|{D}, \theta)p(\theta|{D})d\theta\approx \frac{1}{M}\sum_{j=1}^M p(y_x|{D}, \theta_j),  \theta_j \sim p(\theta_j|{D}),
> > > $$
> > > is a Gaussian Mixture Model (GMM).
> > >
> > > Within BAL and BO, the fully Bayesian treatment is often extended to involve the acquisition function, such that the acquisition function $\alpha$ is computed as an expectation over the hyperparameters [Osborne 2010, Snoek et. al., 2012]
> > >  $$
> > >  \alpha(x|\mathcal{D}) = E_\theta[\alpha(x|\theta, \mathcal{D})] \approx \frac{1}{M}\sum_{j=1}^M \alpha(x|\theta, \mathcal{D}),  \theta{}_j \sim p(\theta{}|\mathcal{D}).
> > >  $$
> > >
> > > This is also the definition of fully Bayesian treatment considered in this work.
> > >
> > > ### 5. Complexity of method
> > >
> > > __Line 193:__ Add a new paragraph inside the SCoreBO Sec. 3.2:
> > >
> > > The conditioning on the fantasized data point involves a rank-1 update of $\mathcal{O}(n^2)$ of the GP for each draw. As such, the complexity of constructing the acquisition functions is $\mathcal{O}(MNn^2)$ for $M$ models, $N$ optima per model and $n$ data points. We utilize NUTS [Hoffman & Gelman, 2014] for the MCMC involved with the fully Bayesian treatment, at a cost of $\mathcal{O}(Dn^3)$ per sample.
> > >
> > > ### 6. Explicit limitations section
> > > __Line 321:__
> > >
> > > _Replace:_
> > >
> > > Moreover, the potential downside of self-correction is displayed when the model structure does not support the task at hand, or when self-correction is not required to solve the task.
> > >
> > > _with:_
> > >
> > > __Limitations__
> > >
> > > SCoreBO displays the ability to increase optimization efficiency on complex tasks that necessitate accurate modeling. However, SCoreBO’s efficiency is ultimately contingent on the intrinsic ability of the GP to model the task at hand, i.e., whether the black-box function is a draw from the class of functions defined by the chosen kernel. Appendix C3 demonstrates this issue for the Rosenbrock (4D) function, where SCoreBO performs worse relative to other acquisition functions. On Rosenbrock, the hyperparameter values increase over time instead of converge, which suggests that the latent function is not part of the model class. Thus, the self-correction effort of SCoreBO is not helpful towards efficient optimization. Moreover, increasing the model capacity, such as in Sec. 4.4, comes with increasing resources allocated towards self-correction. In highly-constrained-budget applications, such resource allocation may not yield the best result, especially if increased model complexity is unwarranted. This is evident from the synthetic AddGP tasks, where despite identifying the additive components more efficiently, SCoreBO does not provide substantial performance gains over EI. Lastly, SCoreBO’s reliance on fully Bayesian hyperparameter treatment makes it substantially more computationally demanding than MAP-based alternatives, limiting its use in high-throughput applications.

---

### Decision · Program_Chairs · 2023-09-21

**Decision:**

Accept (poster)

**Comment:**

This paper proposes new acquisition functions for Bayesian active learning and Bayesian optimization based on disagreement over posterior hyperparameter samples.  Comprehensive experimental results are shown, where the proposed methods perform favorably against baselines.

Most reviewers acknowledged significant contributions but originally raised major concerns:

- Inadequate literature review
- Novelty overstated
- Lack of baseline experiments
- Unclear motivation of ScoreBO.  Advantage over JES?
- Not many real-world data experiments

Through the extensive discussion between the reviewers and the authors, the major concerns have been (not fully but) mostly addressed.  The authors are expected to revise the paper as they promised, which will significantly improve the paper.